# Par complex cluster formation mediated by phase separation

Ziheng Liu[1,6], Ying Yang[2,6], Aihong Gu[1,6], Jiawen Xu[1], Ying Mao[1], Haojie Lu[1], Weiguo Hu [1,3], Qun-Ying Lei [3], Zhouhua Li [4], Mingjie Zhang [5], Yu Cai [2✉] & Wenyu Wen [1✉]

The evolutionarily conserved Par3/Par6/aPKC complex regulates the polarity establishment of diverse cell types and distinct polarity-driven functions. However, how the Par complex is concentrated beneath the membrane to initiate cell polarization remains unclear. Here we show that the Par complex exhibits cell cycle-dependent condensation in *Drosophila* neuroblasts, driven by liquid–liquid phase separation. The open conformation of Par3 undergoes autonomous phase separation likely due to its NTD-mediated oligomerization. Par6, via C-terminal tail binding to Par3 PDZ3, can be enriched to Par3 condensates and in return dramatically promote Par3 phase separation. aPKC can also be concentrated to the Par3N/Par6 condensates as a client. Interestingly, activated aPKC can disperse the Par3/Par6 condensates via phosphorylation of Par3. Perturbations of Par3/Par6 phase separation impair the establishment of apical–basal polarity during neuroblast asymmetric divisions and lead to defective lineage development. We propose that phase separation may be a common mechanism for localized cortical condensation of cell polarity complexes.

[1] Department of Neurosurgery, Huashan Hospital, the Shanghai Key Laboratory of Medical Epigenetics, the International Co-laboratory of Medical Epigenetics and Metabolism, Ministry of Science and Technology, Institutes of Biomedical Sciences, State Key Laboratory of Medical Neurobiology and MOE Frontiers Center for Brain Science, School of Basic Medical Sciences, Fudan University, Shanghai 200032, China. [2] Temasek Life Sciences Laboratory, Department of Biological Sciences, National University of Singapore, Singapore 117604, Singapore. [3] Fudan University Shanghai Cancer Center and Cancer Metabolism Laboratory, Fudan University, Shanghai 200032, China. [4] College of Life Sciences, Capital Normal University, Beijing 100048, China. [5] Division of Life Science, State Key Laboratory of Molecular Neuroscience, Hong Kong University of Science and Technology, Clear Water Bay, Kowloon, Hong Kong, China. [6] These authors contributed equally: Ziheng Liu, Ying Yang, Aihong Gu. ✉email: wywen@fudan.edu.cn; caiyu@tll.org.sg

Cell polarity, the morphological and molecular asymmetries of cells, is essential for diverse processes in metazoan cells[1,2]. A hallmark of cell polarization is local concentration of specific protein complexes at restrictive membrane regions[3,4]. Among them, the Par (partitioning defective) complex, the first set of genes identified in polarization of *Caenorhabditis elegans* embryos during development, are evolutionarily conserved master polarity determinants from worms to mammals[5,6]. The Par complex plays indispensable roles in diverse polarity-related contexts, such as asymmetric cell division (ACD)[7–9], establishment of apical–basal polarity in epithelial cells[3], oriented cell migration[10], and neuronal polarization[11]. Dysfunction of the Par complex leads to developmental defects, tumorigenesis, and even lethality of animals[12].

The Par complex proteins, including Par3 (Bazooka, Baz in *Drosophila*), Par6, and atypical protein kinase (aPKC), are multi-domain proteins capable of binding to each other and a diverse range of other cell polarity-regulating proteins[13]. Par3 has three PDZ (PSD-95, DLG, and ZO-1) domains that mediate protein–protein interactions[14]. How Par3 interacts with Par6 remains controversial. Mammalian Par3 was reported to interact with Par6 through its PDZ domains[15,16], though it is unclear through which PDZ domain Par3 binds to Par6. A recent study showed that both PDZ1 and PDZ3 of Baz weakly bind to a PDZ-binding motif (PBM) of *Drosophila* Par6 (with a dissociation constant > 50 μM)[17]. aPKC, which forms a stable subcomplex with Par6 through their PB1 domains[18], binds to Par3 conserved region 3 (CR3) through its kinase domain, and this inhibitory interaction keeps aPKC in a stable Par complex for the establishment of cell polarity[19]. Activation of aPKC through other regulators (e.g., Aurora-A and Cdc42) leads to the phosphorylation of Par3 CR3 and its subsequent dissociation from Par6/aPKC (ref. [20]). These specific interactions ensure the spatiotemporal localization of the Par proteins at restricted membrane domains to orchestrate cell polarization in different developmental stages and different tissues.

In the past decades, the basic principles of the Par complex assembly and its functions in cell polarity in diverse cell types have been reasonably well established[2–4,8]. However, how are the Par proteins themselves recruited and highly concentrated at very restricted membrane domains to set up the polarity remains unclear. Taking the ACD process of *Drosophila* neuroblasts (NBs) as an example, at the onset of mitosis, the uniformly distributed Baz/Par6/aPKC proteins are gradually concentrated and form a crescent on the apical cortex, whereas cell fate determinants and their adaptor proteins, including the Numb/Pon (Partner of Numb) complex and the Prospero/Miranda (Mira) complex, form crescents on the basal cortex, thus establishing the apical–basal polarity[21–27]. During cell polarization in *C. elegans* zygotes, a similar Par crescent is observed on the anterior cortex[5,7]. Recent studies on *Drosophila* epithelia development and *C. elegans* embryonic polarization demonstrated that such enriched Par crescent is actually an assembly of numerous micrometer-sized Par clusters, and formation of Par clusters requires the oligomerization of Par3 through its N-terminal domain (NTD)[28–32], which self-associates to form helical filaments[33]. However, there is still a significant gap in understanding how the Par3 filaments in vitro establish the dynamic Par clusters in vivo that are capable to fuse with each other into larger ones[30,34]. Interestingly, the Par proteins in the cortical clusters and other polarity complexes in the crescents are highly dynamic, and can rapidly exchange with the proteins in cytoplasm[30–32,35–38]. It is not clear how these inner membrane-attached polarity complexes maintain highly localized concentration in context of the sharp concentration gradients between cell cortex and cytoplasm.

In this work, we uncover that endogenous Par proteins form discrete puncta-shaped condensates during the establishment of

apical–basal polarity in *Drosophila* NBs. Mammalian Par3 PDZ3 specifically recognizes Par6 PBM, and the Par3/Par6 interaction can be significantly enhanced by Par3 NTD and Par6 PB1 through their oligomerization. Such multivalent interaction between Par3 and Par6 leads to the formation of self-organized, highly condensed, and dynamic droplets/puncta through liquid–liquid phase separation (LLPS) both in vitro and in vivo. Mutations that impair the LLPS of the Par complex led to defective assembly of the apical Par complex crescent during *Drosophila* NB asymmetric divisions, and consequently resulted in ACD defects and defective cell lineage. Thus, the local condensation of the Par complex during cell polarization is driven by LLPS mediated by their multivalent interactions.

## Results

**Par proteins exhibit cell cycle-dependent clustering in NBs.** During the ACD of *Drosophila* NBs, Baz, Par6, and aPKC are found to form a co-localized, condensed crescent on the apical membrane at metaphase[25–27]. However, careful analyses of microscopy images of dividing NBs showed that endogenous Baz, Par6, and aPKC displayed cell cycle-dependent puncta formation on the apical cortex (Fig. 1), just like their *C. elegans* counterparts during embryonic polarization[30,31]. From prophase, Par proteins emerged as scattered puncta on the apical cortex, and these puncta concentrated to form a condensed crescent (from the apical–basal view) on the apical membrane at metaphase. Interestingly, when viewed from the lateral apical direction, the dim Par puncta grew into larger and brighter ones from prophase to metaphase, assembling into a highly concentrated cluster around the apical pole. From anaphase, the highly condensed puncta cluster began to disassemble into scattered small puncta, and finally the Par protein signal was distributed on the whole cortex of the apical daughter cell (Fig. 1).

The condensed Par puncta cluster was sensitive to 1,6-hexanediol treatment, a molecule known to disturb hydrophobic interaction-induced phase separation assemblies both in vitro and in vivo[39,40]. Larval brain treated with 1,6-hexanediol, exhibited defective localization of these endogenous apical Par components, as well as the basal cortex component Mira, in a concentration-dependent manner (Supplementary Fig. 1a–c). In these NBs, Par proteins and Mira were partially localized with certain cytoplasmic distribution when treated with 5% 1,6-hexanediol, and fully diffused in cytoplasm with 10% 1,6-hexanediol. Importantly, removal of 1,6-hexanediol restored the crescent formation of Baz and Par6, as well as aPKC and Mira in dividing NBs, indicating that the cell cycle-dependent apical condensation of the Par complex is a reversible process and is likely driven by phase separation.

**Par6β promotes Par3 puncta formation in living cells.** The fact that the membrane-attached, highly concentrated Par3/Par6/aPKC assemblies are in open contact and fast protein equilibrium with the cytoplasm[30–32,35] led us to explore whether Par proteins can spontaneously form condensates in the aqueous cytoplasm. We overexpressed mammalian Par3, Par6β, or PKCι (aPKC in mammals) individually, or co-expressed multiple Par components in COS7 cells. To our surprise, COS7 cells transfected with green fluorescent protein (GFP)- or mCherry-tagged mammalian Par3, Par6β, or PKCι alone showed diffused cytoplasmic localization of these proteins (Fig. 2a). So did those co-expressed with two components of the complex (Fig. 2b). It was suggested that Par3 adopts an auto-inhibited closed conformation, with its C-terminal conserved 4N12 region packing with its N-terminal part covering NTD and PDZ1–3, and thus inhibiting its microtubule-binding property[41]. Interestingly, when ectopically expressed in COS7 cells, in addition to fibrillary structures[41], the Par3 4N12 deletion mutant

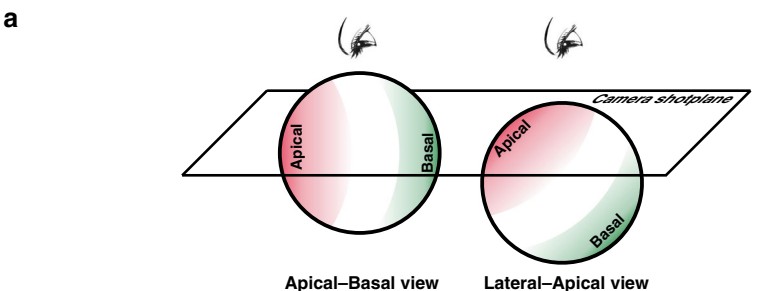

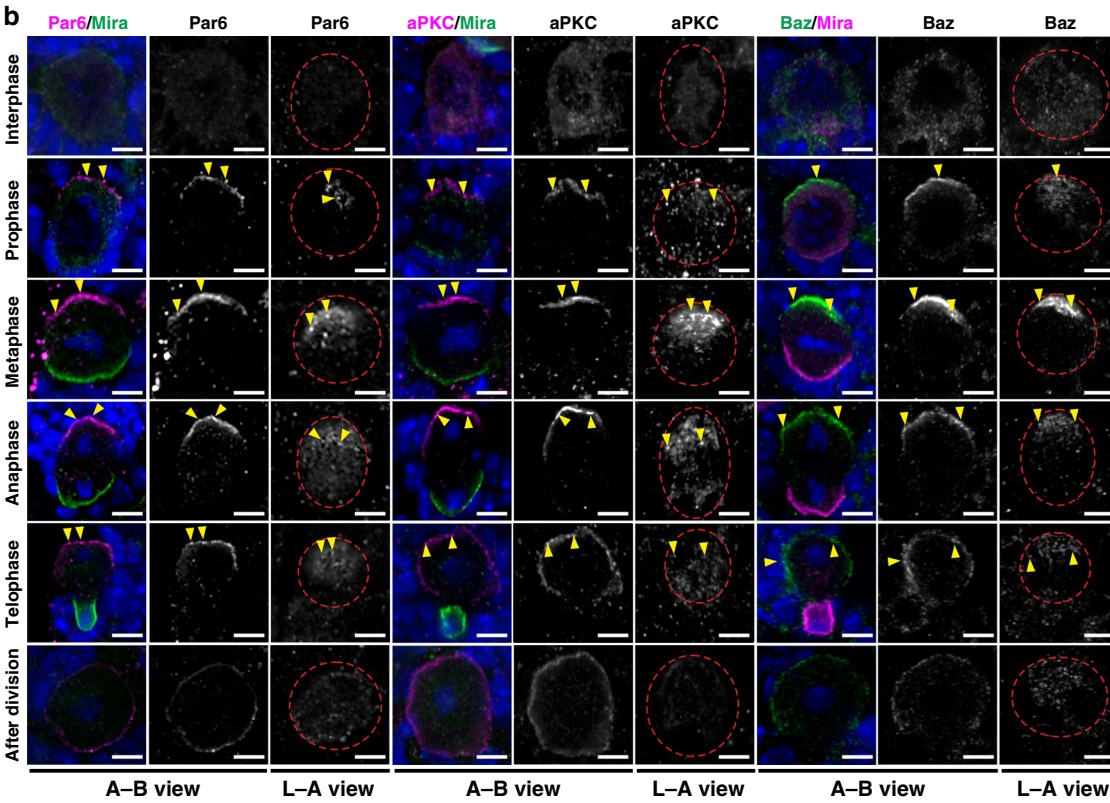

**Fig. 1 Endogenous Par proteins form condensed puncta in *Drosophila* larval NBs in a cell cycle-dependent manner. a** Schematic diagram showing the view planes. **b** Representative image of endogenous Baz, Par6, aPKC, and Mira at different cell cycle stages ($n = 10$ neuroblasts collected from ten larval brains over three independent experiments). ToPro-3 in blue. Yellow arrowheads indicate condensed puncta on the apical cortex. Scale bars, 5 μm. Experiments were performed three times independently with similar results.

(Par3 Δ4N12; Supplementary Table 1) also formed scattered dim puncta in some cells (Fig. 2a). A truncated form of Par3 containing NTD and PDZ1–3 (referred to as Par3N) showed a similar dim puncta distribution as Par3 Δ4N12 did (Figs. 2a and 3a), indicating that domains within Par3N possess the ability to spontaneously form concentrated puncta in the cytoplasm.

We further found that co-expression of Par6β together with Par3N dramatically promoted the puncta formation. GFP-Par3N and mCherry-Par6β spontaneously assembled into highly concentrated bright puncta in cytoplasm showing both GFP and mCherry signals (Fig. 2b). In sharp contrast, co-expression of PKCι with Par3N did not show such promotion effect. As no puncta was observed when co-expressing Par6β with full-length Par3 (Fig. 2b), we proposed that Par6β only promotes the puncta formation of Par3 in its open state. The Par3N and Par6β-enriched puncta were originally small and gradually fused into larger ones in a time-dependent manner (Fig. 2c, Supplementary Movie 1). Importantly, fluorescence recovery after photobleaching (FRAP) analysis revealed that both Par3N and Par6β in the condensed puncta rapidly exchanged with the proteins in the surrounding cytoplasm (~75% recovery with a halftime ~10 s; Fig. 2d–f, Supplementary Movies 2 and 3), reminiscent of the observations showing that Par proteins are in fast equilibrium between the anterior cortex and the cytoplasm during *C.elegans* embryonic polarization[30–32,35]. This observation implies that the Par3N and Par6β-enriched puncta formed in cytoplasm may possess some of the properties of Par condensates beneath the membrane in polarized cells.

**Phase separation of Par3N with Par6β in vitro**. Next, we asked whether Par proteins undergo phase separation in vitro. Due to the poor protein behavior of Par3 Δ4N12, we chose Par3N for the LLPS assay. When assessing iFluor488-labeled Par3N with or without Par6β under fluorescence microscope, we observed numerous small spherical droplets with various diameters, a phenomenon characteristic of LLPS (Fig. 3a). The phase separation level of Par3N or Par3N/Par6β proteins was correlated to its or their concentration (s). When the protein concentration increased, the amount and the size of liquid droplets increased, and the Par3N/Par6β complex formed droplets with much larger amount and size than Par3N

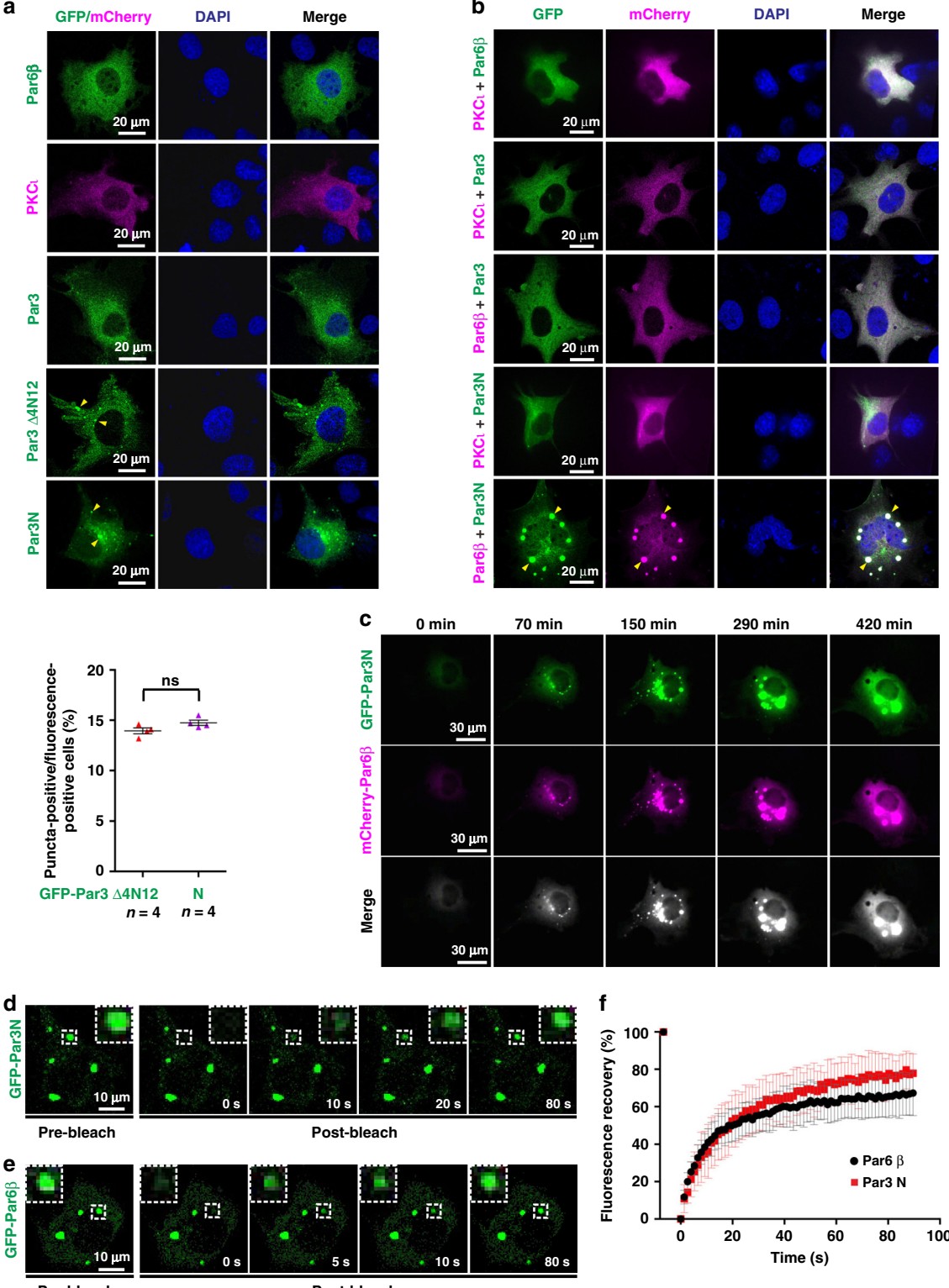

itself at the same protein concentration (Fig. 3a, b). The distributions of Par3N or Par3N/Par6β complex between the aqueous phase and the condensed droplets were quantified by a sedimentation-based assay. The LLPS level of Par3N/Par6β complex was constantly higher than that of Par3N alone at the same concentration before reaching saturation, with obvious LLPS at very low concentration in vitro (~0.5 μM; Fig. 3c, d), implying an important role of Par6β in promoting LLPS of Par3N.

Obvious fusion of the iFluor488-Par3N droplets could be observed under differential interference contrast (DIC) microscope at a protein concentration of 25 μM (Fig. 3e, Supplementary Movie 4). When mixed with Cy3-tagged Par6β at 1:1 molar ratio, every droplet was enriched with both Par6β and Par3N (Fig. 3f, Supplementary Movie 5). In sharp contrast, isolated solutions of Par6β remained clear at the same concentration (Supplementary Fig. 2a). Par3N either in the isolated or Par6β-

**Fig. 2 Par proteins spontaneously form condensed and dynamic puncta in COS7 cells. a**, **b** Representative images showing subcellular localizations of GFP- or mCherry-tagged Par complex components, including various Par3 fragments (full length, Δ4N12, and Par3N), Par6β, and PKCι, when expressed individually **a** or mutually **b** in COS7 cells. Yellow arrowheads indicate condensed puncta in the cytoplasm. Nuclei were stained by DAPI. The lower panel of **a** is the statistical data for the puncta formation of GPF-tagged Par3 Δ4N12 and Par3N. $n$ = number of independent experimental cell culture batches, with 800 cells counted for each batch. Specimens' statistics are presented as mean ± SEM; ns, not significant, using one-way ANOVA with Tukey's multiple comparison test. **c** Representative time-lapse images showing that the GFP-Par3N and mCherry-Par6β-positive puncta undergo time-dependent fusion. **d** Representative time-lapse FRAP images showing that GFP-Par3N signal within the GFP-Par3N/Flag-Par6β condensed puncta recovered within a few minutes. **e** Representative time-lapse FRAP images of GFP-Par6β in the HA-Par3N/GFP-Par6β condensed puncta. **f** Statistical data for **d** and **e**. The red curve represents the averaged FRAP data of 20 puncta from 14 cells. The black curve represents the averaged FRAP data of 25 puncta from 13 cells. Time 0 refers to the time point of the photobleaching pulse. Experiments were performed three times independently with similar results. All data are represented as mean ± SD. All the constructs are listed in Supplementary Table 1. Source data are provided as a Source data file.

bound droplets exchanged rapidly between the condensed phase and the surrounding aqueous solution (Fig. 3g, Supplementary Movies 6 and 7). Those Par3N/Par6β droplets transitioned into gel-like structures over time, and the droplets rarely fused together after 20 min (Supplementary Fig. 2b).

**Par3 PDZ3 recognizes Par6β PBM**. Next, we wanted to find out how Par6β promotes Par3N phase separation. Cell lysate glutathione S-transferase (GST) pull-down assay confirmed the interaction between Par6β and Par3N (Fig. 4a–c). Par6β selectively bound to Par3N PDZ3 though GST-Par3N pulled down more Par6β than GST-Par3 PDZ3 did (Fig. 4b), likely due to the oligomerization of NTD (ref. [33]), as it has been shown that Par6 preferentially binds to Par3 oligomers but not monomers[32]. Besides, Par3N selectively interacted with Par6β PBM (Fig. 4c). We confirmed the interaction between Par3 PDZ3 and Par6β PBM with a dissociation constant of ~1 μM, which was ~50-fold stronger than that in *Drosophila*[17], whereas mammalian Par3 PDZ1 hardly bound to Par6β PBM (Fig. 4d).

The crystal structure of Par3 PDZ3 in complex with a Par6β PBM peptide (Fig. 4a) revealed that the last three residues of Par6β PBM ($^{-2}$ITL$^0$) binds to the αB–βB groove of PDZ3 using a classical PDZ binding mode (Fig. 4e, f). The carboxyl terminal of L(0) from Par6β PBM forms extensive hydrogen bonds with the canonical "G$^{600}$LG$^{602}$F" loop of Par3 PDZ3. The sidechain of I (−2) from Par6β PBM inserts into a shallow hydrophobic pocket formed by the αB–βB groove of Par3 PDZ3. In line with the structure, the G600,602 A mutation of Par3 PDZ3 dramatically weakened the PDZ3–PBM interaction ~20-fold (Fig. 4d). We further found that addition of 1,6-hexanediol impaired the hydrophobic interaction between Par3 PDZ3 and Par6β PBM (Supplementary Fig. 2c). In line with this result, addition of 1,6-hexanediol led to the dispersion of the preformed Par3N/Par6β droplets (Supplementary Fig. 2d, e, Supplementary Movie 8), reminiscent the finding that it disturbed the apical localization of Par complex in dividing *Drosophila* NBs (Supplementary Fig. 1).

It is noted that Par6β Crib-PDZ slightly enhanced the binding between PBM and Par3N, and addition of PB1 dramatically strengthened the interaction (Fig. 4c). We observed that the recombinant Par6 PB1 was prone to aggregation, with a small elution volume in the size-exclusion chromatography (SEC) assay (Supplementary Fig. 3a). Different from Par3 NTD whose aggregation was sensitive to salt concentration[33], high salt concentration up to 2 M NaCl had negligible impact on Par6 PB1 oligomerization (Supplementary Fig. 3b). In line with the SEC result, the full-length Par6β could self-associate through its PB1 domain (Supplementary Fig. 3c, d).

Overall, Par3N specifically binds to Par6β via the PDZ3–PBM recognition, and both Par3 NTD and Par6β PB1 enhance the interaction likely through oligomerization-induced avidity increase.

**Multivalency-dependent LLPS of Par3N with Par6β**. We then investigated the specific role of the direct Par3N–Par6β interaction or oligomerization of each protein in promoting Par3N/Par6β phase separation, at a concentration relatively low (~3 μM) but with nearly saturating LLPS capacity (Fig. 3d). Consistent with the above biochemical data (Fig. 4), deletion of PDZ3 (ΔPDZ3) but not PDZ1 (ΔPDZ1) from Par3N effectively impaired LLPS of the Par3N/Par6β complex (Fig. 5a). Removing NTD (ΔNTD) from Par3N significantly reduced phase separation efficiency of the Par3N/Par6β complex, as well as that of Par3N alone, demonstrating the important role of Par3 NTD self-association in increasing valency of Par3N for higher order architecture assembly. The LLPS of Par3N/Par6β was sensitive to salt concentration (Supplementary Fig. 3e), as the oligomerization of Par3 NTD is charge–charge interaction dependent[33].

On the other hand, deletion of PBM (ΔPBM) not only impaired LLPS of Par6β, but also reduced its promotion effect on LLPS of Par3N (Fig. 5b). Deletion of PB1 (ΔPB1) from Par6β also dramatically decreased the condense phase of Par6β (Fig. 5b). Moreover, when we replaced PB1 of Par6β with another self-associating domain p62 PB1 (referred to as Par6β chimera)[42], the phase separation efficiency of the Par3N/Par6β chimera can be restored to the similar extent as that of the Par3N/Par6β complex (Supplementary Fig. 3f), demonstrating the critical role of Par6β PB1 oligomerization in promoting LLPS. Due to the role of Par6β Crib-PDZ in increasing Par6β–Par3N interaction (Fig. 3c), deletion of Par6β Crib-PDZ (ΔCrib-PDZ) slightly reduced LLPS efficiency of the Par3N/Par6β complex (Fig. 5b).

Together, Par3N undergoes weak phase separation in vitro, at least partially dependent on its NTD-mediated oligomerization. Par6β promotes Par3N LLPS, relying on the specific interaction between Par3 PDZ3 and Par6β PBM, as well as the high valency of both proteins.

**Par puncta in living cells are phase separated droplets**. Next, we wanted to test whether the multivalency-dependent LLPS of the Par3N/Par6β complex in solution accounts for the puncta formation in living cells (Fig. 2). Nicely correlated with in vitro phase separation results (Fig. 5a), deletion of Par3 PDZ1, PDZ2, or even PDZ12 had minor effects on the puncta formation of the co-expressed Par3N and Par6β, whereas deletion of the Par6β-binding Par3 PDZ3 led to a dramatic decrease of the puncta number (Fig. 5c, Supplementary Fig. 4a). When introducing a V13D, D70K mutation in Par3 NTD (referred to as NTDmu), which disrupts its oligomerization[33], only very few puncta could be detected. Similarly, fewer puncta were observed when Par3N was co-expressed with Par6β mutants with PB1 or Crib-PDZ deletions, and deletion of Par6β PBM eliminated the puncta formation (Fig. 5d, Supplementary Fig. 4b).

There are three isoforms of Par6 (Par6α, Par6β, and Par6γ) in mammals, which share ~60% sequence similarity (Fig. 4a) and exert distinct functions in cell polarization, e.g., tight junction

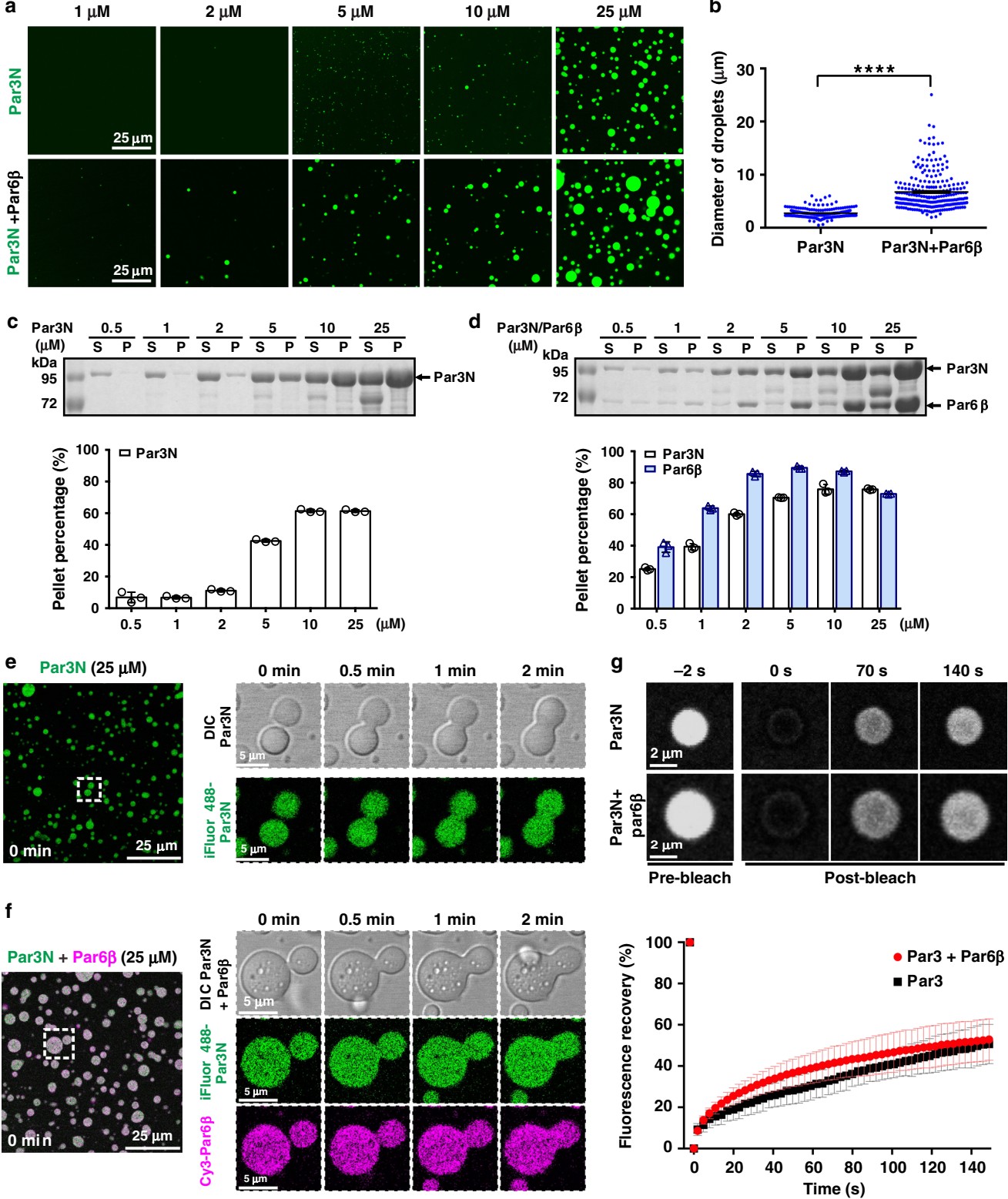

formation[43]. As a key driving factor for the specific interaction and LLPS of Par3N and Par6β, Par6β PBM is completely conserved in Par6γ but varies in Par6α (Fig. 4a). As expected, Par6γ PBM but not Par6α PBM bound to Par3 PDZ3 equally well as that of Par6β PBM (Fig. 4d). Many bright puncta were observed in the cytoplasm when co-expressing Par3N with Par6γ, and deletion of Par6γ PBM eliminated the puncta formation (Fig. 5d, Supplementary Fig. 4b). In sharp contrast, no puncta

were observed in Par3N and Par6α co-expressed cells. We then constructed a Par6α chimera mutant, with its PBM replaced by the Par6β PBM, and the puncta formation property was gained in cells co-expressed with Par3N and the Par6α chimera (Fig. 5d, Supplementary Fig. 4b), further demonstrating the critical role of PBM–PDZ3 recognition in promoting puncta formation of Par3N in living cells. As LLPS of Par3N/Par6β was concentration dependent (Fig. 3), the expression level of each protein in above

**Fig. 3 Par6β promoted LLPS of Par3N in vitro. a** Protein concentration-dependent LLPS of Par3N or Par3N/Par6β complex. Only Par3N was iFluor[TM] 488 labeled. The fluorescence imaging settings were identical for easy comparison. Images were acquired at ~2 min after injecting the mixture into the chamber. **b** Column scatter charts show the droplet size of Par3N (25 μM, n = 250 droplets examined over five independent observation fields) or Par3N/Par6β complex (25 μM, n = 250 droplets examined over five independent observation fields). Data are shown as mean ± SEM. ns, not significant; *$p < 0.05$, **$p < 0.01$, ***$p < 0.001$, and ****$p < 0.0001$ using one-way ANOVA with Tukey's multiple comparison test. **c**, **d** Representative SDS–PAGE analysis and quantification data showing the distribution of proteins between aqueous solution/supernatant (S) and condensed liquid phase/pellet (P) fractions. Par3N and Par6β were mixed at a 1:0 **f** or 1:1 **g** molar ratio at various concentrations. Experiments were performed three times independently with similar results. Data are expressed as mean ± SD. **e**, **f** The time-lapse images showing the localization of iFluor[TM] 488-Par3N **e** or co-localization of iFluor[TM] 488-Par3N and Cy3-Par6β complex **f** in the droplets with enriched concentrations. The enlarged images at right show that small droplets undergo time-dependent coalescence into larger ones. The 0 min images were acquired at ~5 min after injecting the mixture into the chamber. **g** FRAP analysis of iFluor[TM] 488-Par3N droplets in the absence or presence of Par6β in vitro showing the exchange kinetics of the protein in droplets with the surrounding aqueous solution. The curves below represent FRAP recovery curves of iFluor[TM] 488-Par3N (with or without Par6β) by averaging signals of 20 droplets with similar sizes each after photobleaching. Time 0 refers to the time point of the photobleaching pulse. Experiments were performed three times independently with similar results. Data are represented as mean ± SD. Source data are provided as a Source data file.

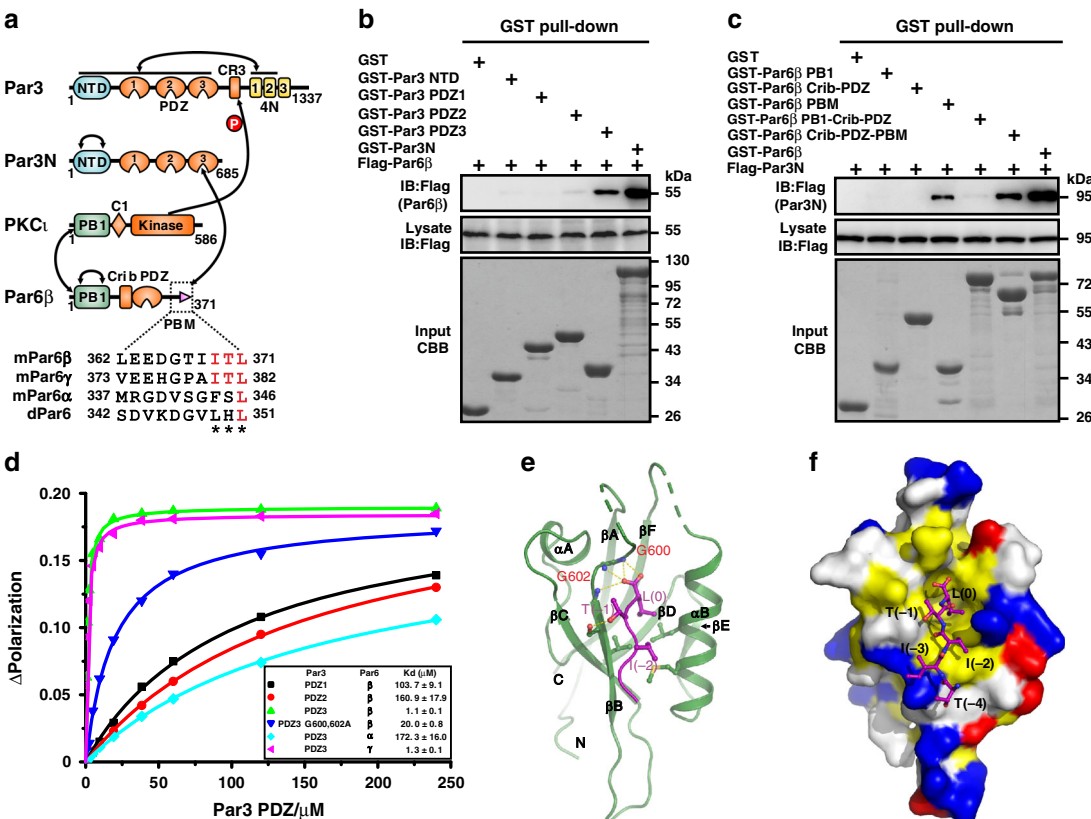

**Fig. 4 Par3 recognizes Par6 via the interaction between Par3 PDZ3 and Par6 PBM. a** Schematic diagrams showing the domain organizations of Par3, Par6β, and PKCι. Amino acid sequences of the PBM from mouse Par6α, Par6β, Par6γ, and *Drosophila* Par6 are present bellow, with the completely conserved residues colored in red. **b** Cell lysate GST pull-down assay of various GST-tagged rat Par3 fragments (NTD, PDZ1, PDZ2, PDZ3, or Par3N) with Flag-Par6β. Par3 PDZ3 specifically bound to Par6β, and the interaction was strongly enhanced by Par3 NTD. **c** Cell lysate GST pull-down assay of various GST-tagged mouse Par6β fragments (PB1, Crib-PDZ, PBM, PB1-Crib-PDZ, Crib-PDZ-PBM, or full-length Par6β) with Flag-Par3N. Par6β PBM specifically bound to Par3, which was sharply enhanced by Par6β PB1. **d** Fluorescence polarization-based measurements of the binding affinities between Par3 PDZs (PDZ1, PDZ2, PDZ3, or PDZ3 G600, 602 A) and various Par6 (Par6α, Par6β, or Par6γ) PBM peptide. **e** Ribbon representation of the Par3 PDZ3 (green)/Par6β PBM (purple) complex. Key residues involved in Par3/Par6 binding are shown as a ball-and-stick model. **f** Combined surface (Par3 PDZ3) and stick (Par6β PBM) diagram of the complex. In the surface map, hydrophobic residues are colored in yellow, positively charged residues in blue, negatively charged residues in red, and other residues in white. Experiments were performed three times independently with similar results. All the constructs are listed in Supplementary Table 1. Source data are provided as a Source data file.

puncta assay was analyzed and adjusted to make sure that the imaging-based puncta formation assay of different constructs are directly comparable (Supplementary Fig. 4c, d).

The essential role of Par3 in promoting Par complex clustering and eventually the establishment of cell polarity is thought to be mediated by its NTD (refs. [28–32]). Par3 NTD can self-associate to

form helical filaments in vitro[33], which may act as a central organizer for Par complex clustering in vivo[34]. An intriguing question is, does NTD-mediated Par clustering rely on its self-association nature, or the oligomerization-dependent LLPS behavior? To answer this question, we replaced Par3 NTD with different fragments of the unstructured low-complexity domain

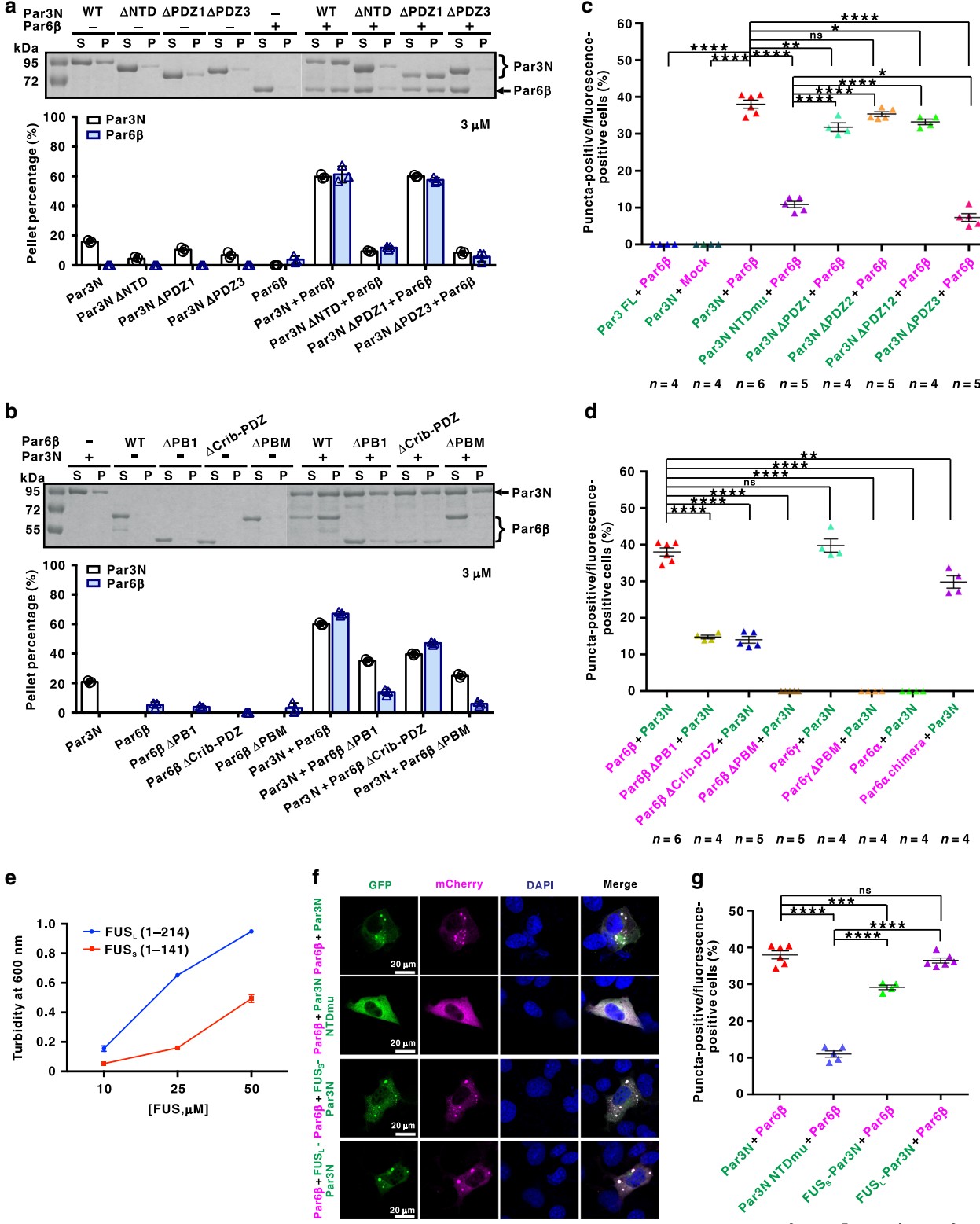

(LCD) of fused in sarcoma (FUS), the driving factor for FUS LLPS (refs. [44,45]). The LLPS level of the longer fragment $FUS_L$ (aa 1–214) is much higher than that of the shorter $FUS_S$ (aa 1–141; Fig. 5e). Accordingly, $FUS_S$-Par3N chimera partially restored but $FUS_L$-Par3N chimera essentially restored the puncta formation property of Par3N when co-expressed with Par6β (Fig. 5f, g), indicating that Par3-mediated Par complex clustering is most likely driven by its LLPS.

**PKCι activity may regulate LLPS of the Par complex.** As it has been suggested that Par6 and aPKC form constitutive heterodimers[18,31], we next wondered whether aPKC, the known kinase in the Par complex, might participate in the phase separation of Par3 and Par6β. Under fluorescence microscopy, iFluor405-labeled PKCι PB1 was well co-localized with the condensed Par3N (without PKCι phosphorylation site)/Par6β droplets (Fig. 6a). Sedimentation assay further showed that though

**Fig. 5 Multivalent and specific protein–protein interactions drive LLPS of Par3N/Par6β complex. a** Sedimentation assay of various Par3N fragments (WT, ΔNTD, ΔPDZ1, and ΔPDZ3), Par6β, or both proteins mixed at a 1:1 molar ratio at 3 μM. The NTD and PDZ3 domains of Par3 are critical for the phase separation of the Par3N/Par6β complex. **b** Sedimentation assay of Par3N, various Par6β fragments (WT, ΔPB1, ΔCrib-PDZ, and ΔPBM), or both proteins mixed at a 1:1 molar ratio at 3 μM. The PB1 and PBM of Par6β are critical for the phase separation of the Par3N/Par6β complex. All statistic data in **a** and **b** represent the results from three independent batches of experiments and are expressed as mean ± SD. **c** Puncta formation summary for co-expression of GFP-Par3 WT or various mutants (Par3N, Par3N NTDmu, Par3N ΔPDZ1, Par3N ΔPDZ2, Par3N ΔPDZ12, and Par3N ΔPDZ3) with mCherry-Par6β or mCherry vector (Mock) in COS7 cells. Statistical data for Supplementary Fig. 4a. **d** Puncta formation summary for co-expression of GFP-Par3N with various mCherry-Par6 fragments (Par6β, Par6β ΔPB1, Par6β ΔCrib-PDZ, Par6β ΔPBM, Par6γ, Par6γ ΔPBM, Par6α, and Par6α chimera) in COS7 cells. Statistical data for Supplementary Fig. 4b. **e** Phase separation/turbidity diagram for FUS$_L$ (1–214) and FUS$_S$ (1–141). Error bars, mean ± SEM, n = 5. **f** Representative images showing expression of GFP-Par3N WT or various mutants (Par3N NTDmu, FUS$_L$-Par3N with NTD replaced by FUS$_L$, FUS$_S$-Par3N with NTD replaced by FUS$_S$) with mCherry-Par6β in COS7 cells. Nuclei were stained by DAPI. **g** Statistical data for **f**. n = number of independent experimental cell culture batches, with 800 cells counted for each batch. Specimens' statistics are presented as mean ± SEM; ns, not significant; *p < 0.05, **p < 0.01, ***p < 0.001, and ****p < 0.0001 using one-way ANOVA with Tukey's multiple comparison test. All the constructs are listed in Supplementary Table 1. Source data are provided as a Source data file.

PKCι PB1 could be recruited into the Par3N/Par6β liquid phase, it did not enhance or decrease the LLPS extent of Par3N/Par6β proteins (Fig. 6b). Similarly, the full-length PKCι was enriched and co-localized within the Par3N/Par6β puncta in COS7 cells (Supplementary Fig. 5a), and co-expression of PKCι had negligible impact on Par3N/Par6β puncta formation (Fig. 6c), suggesting that PKCι is likely recruited into the Par3N/Par6β condensed phase as a client.

aPKC can phosphorylate Par3 in its CR3 region (Fig. 4a), leading to the release of Par3 from the Par6/aPKC complex[19,46]. To probe the potential impact of aPKC kinase activity on Par3/Par6β LLPS, we co-expressed mCherry-Par6β with GFP-Par3 Δ4N12 (the open form of Par3 containing the aPKC phosphorylation sites) wild-type protein (WT) or the aPKC-mediated phospho-mimetic S827,829E mutant in COS7 cells. Fewer puncta were observed when Par6β was co-expressed with Par3 Δ4N12 S827,829E when compared with Par3 Δ4N12 (Fig. 6d, Supplementary Fig. 5b), indicating that PKCι-mediated Par3 phosphorylation at CR3 somehow interfered the Par3 Δ4N12/Par6β puncta formation. Then, we co-expressed GFP-Par3 Δ4N12 with Flag-Par6β and mCherry-PKCι WT, and various mutants in COS7 cells. To our surprise, PKCι WT, the kinase-dead K273R mutant[47], and the proposed constitutively active A120E mutant[48] were all enriched and co-localized within the Par3 Δ4N12/Par6β puncta (Fig. 6e, Supplementary Fig. 5c). Expression levels of each protein in above puncta assay were confirmed to be comparable (Supplementary Fig. 5d).

It has been proposed that association with Par3 and Par6 keeps aPKC in the inactive quiescent state in the Par complex[19,31], and this inhibitory interaction can be disrupted by other regulatory factors in a spatiotemporal manner[20,31]. We then asked whether the recruitment of aPKC into Par3/Par6 condensates might keep aPKC in the inactive state. We conducted a sedimentation assay of Par3 1–854 (containing the aPKC phosphorylation sites) and Par6β with or without incubation of Flag-PKCι purified from HEK293 cells. Both PKCι WT and A120E efficiently phosphorylated Par3 1–854 (indicated by the upshifted phos-tag band) in the supernatant fraction, but not in the Par3/Par6β/PKCι condensates (Fig. 6f), indicating that though PKCι could be recruited into the Par3/Par6β condensates, its activity is suppressed.

In summary, aPKC can be recruited and concentrated into the Par3/Par6β condensates as an inactive client. However, activation of aPKC (e.g., cell cycle-dependent activation) will convert the kinase into an active Par3/Par6β condensate disperser likely via aPKC-mediated phosphorylation on Par3.

**LLPS of Par proteins regulates their condensation in vivo.** Both Par3 and Par6 have evolved into multiple isoforms in mammals with redundant functions. *Drosophila* contains only one copy of *par3* (*bazooka*, *baz*) and *par6* genes, making the organism

desirable for functional studies on Par3/Par6 condensation. Importantly, Par6β PBM is highly conserved in *Drosophila*, e.g., the critical L(−2) of fly Par6 is a small hydrophobic residue, just like I(−2) in Par6β and Par6γ (Fig. 3a), though the binding between Baz PDZ3 and Par6 PBM is relatively weak in flies[17]. Thus, we employed type I NBs in the central brain region of *Drosophila* larvae as an in vivo model. We generated transgenic flies expressing the full-length Flag-tagged WT or LLPS-deficient mutant forms of Baz or Par6, respectively. Expressions of these fragments were further verified (Supplementary Fig. 6).

In a WT background under our experimental condition (see Methods section), *Flag-Baz WT* (n = 11), *Flag-Par6 WT* (n = 20), and *Flag-Baz ΔPDZ2* (n = 19) that had high LLPS ability in vitro formed a crescent on the apical cortex in all or the majority of metaphase NBs (Fig. 7a–d). In contrast, the Par6-binding-deficient *Flag-Baz PDZ3 G634,636 A* (referred to as PDZ3mu, corresponding to rat Par3N PDZ3 G600,602 A; n = 17), and the oligomerization-deficient *Flag-Baz NTD L13D,D68K* (referred to as NTDmu, corresponding to rat Par3 NTD V13D,D70K; n = 23) and *Flag-Par6 ΔPB1* (n = 19) lost their normal apical condensation in the majority of NBs examined. The localization of endogenous apical proteins Par6 and aPKC, and basal proteins Mira and Numb was also disrupted in these WT NBs expressing *Flag-Baz NTDmu* or *Flag-Baz PDZ3mu* (Supplementary Fig. 7a, d), suggesting a dominant-negative effect of both mutants. In contrast, though *Flag-Par6 ΔPB1* diffused in cytoplasm, the localization of endogenous apical Baz and aPKC, and basal Mira and Numb seemed normal (Fig. 7b, Supplementary Fig. 7b, d).

To exclude the influence of endogenous proteins on these ectopically expressed variants, we addressed their localization in *baz* or *par6* mutant NBs. As reported previously[26,27], endogenous Baz (n = 20) was apically localized on the cortex; *Flag-Baz WT* (n = 5), and *Flag-Baz ΔPDZ2* (n = 5) with high LLPS ability also formed apical crescent in the majority of *baz* mutant NBs (Fig. 7e, f). However, majority *baz* mutant NBs expressing LLPS-deficient *Flag-Baz NTDmu* (n = 5) and *PDZ3mu* (n = 8) exhibited diffused cortex and cytoplasm localization (Fig. 7e, f). Similarly, Par6 was correctly localized on the apical cortex either in WT NBs (n = 20) or in *par6* mutant NBs expressing *Flag-Par6 WT* (n = 5; Fig. 7g, h). However, *Flag-Par6 ΔPB1* diffused in the whole cytoplasm in the majority of *par6* mutant NBs (Fig. 7g, h; n = 9).

To further understand the function of these Flag-Baz and Flag-Par6 variants, we investigated whether they could rescue *baz* or *par6* mutant phenotype. In the WT central brain, each type I NB lineage (marked by GFP) contains an average of ~73 cells, including NB, GMCs, and neurons (n = 20); however, baz NBs produced a much smaller lineage containing ~23 cells (n = 20; Fig. 8a, b). While restoring Flag-Baz WT or *Flag-Baz ΔPDZ2* expression in *baz* mutant NBs largely or significantly restored the lineage defect, expression of LLPS-deficient *Flag-Baz NTDmu* and

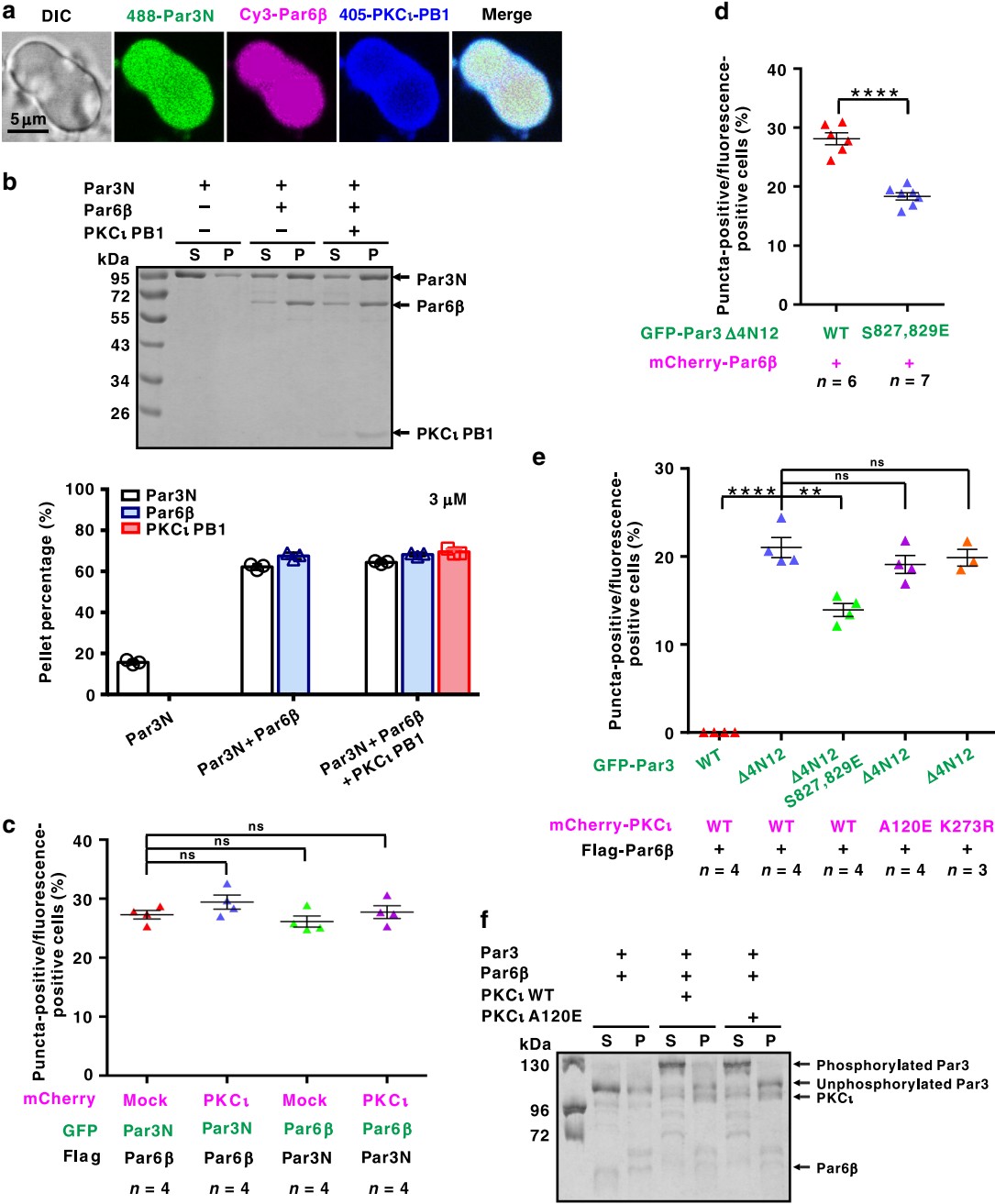

**Fig. 6 PKCι activity may regulate the LLPS of Par3N/Par6β. a** The fluorescent images showing that the co-localization of iFluor^TM 488-Par3N, Cy3-Par6β, and 405-PKCι PB1 in the droplets with enriched concentration. **b** Sedimentation assay and quantification data showing that PKCι PB1 can participate the LLPS of Par3N/Par6β complex as a client with no effect on the extent of LLPS. $n = 3$ biologically independent experiments. Data are represented as mean ± SD. **c** Puncta formation summary for co-localization and condensation of Par3N, Par6β, and PKCι in COS7 cells. Statistical data for Supplementary Fig. 5a. Co-expression of PKCι full-length protein did not enhance or impair LLPS of Par3N/Par6β. **d** Puncta formation summary for co-expression of GFP-Par3 Δ4N12 WT or the aPKC phospho-mimetic S827,829E mutant with mCherry-Par6β in COS7 cells. Statistical data for Supplementary Fig. 5b. **e** Puncta formation summary for co-expression of GFP-Par3 WT, Δ4N12 or the phospho-mimetic Δ4N12 S827,829E with Flag-Par6β, and mCherry-PKCι WT, the constitutively active A120E, or the kinase-dead K273R mutant in COS7 cells. Statistical data for Supplementary Fig. 5c. $n =$ number of independent experimental cell culture batches, with 800 **c**, **d** or 600 **e** cells counted for each batch. Specimens' statistics are presented as mean ± SEM; ns, not significant; *$p < 0.05$, **$p < 0.01$, ***$p < 0.001$, and ****$p < 0.0001$ using one-way ANOVA with Tukey's multiple comparison test. **f** Sedimentation assay showing the distribution of proteins (Par3 1–854, Par6β, PKCι WT, or A120E mutant) between aqueous solution/supernatant (S) and condensed liquid phase/pellet (P) fractions. Both PKCι WT and A120E mutant phosphorylated Par3 1–854 in the supernatant fraction but not in the pellet fraction. The band of phosphorylated Par3 1–854 was resolved by Phos-tag PAGE. Experiments were performed three times independently with similar results. Source data are provided as a Source data file.

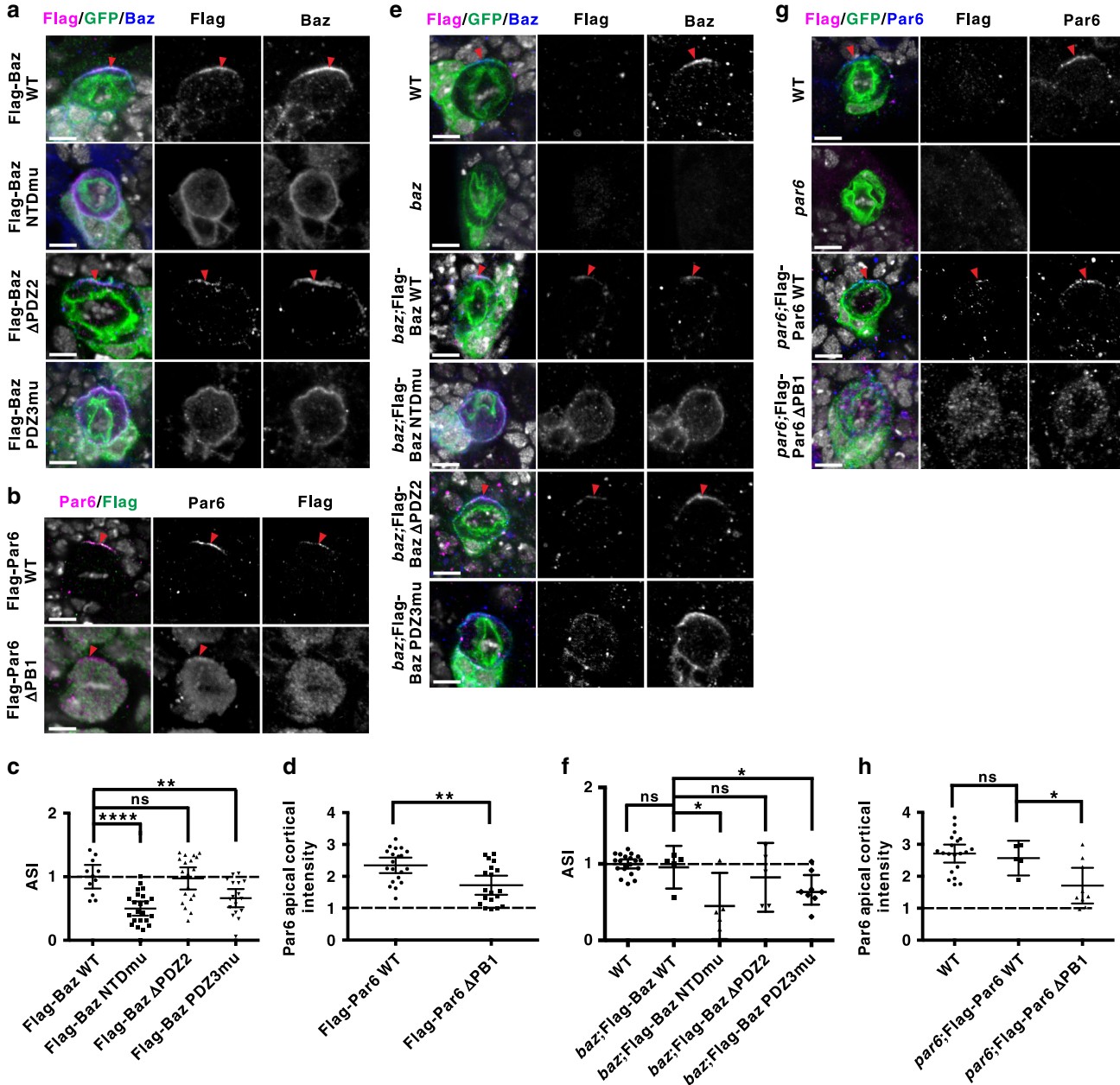

**Fig. 7 LLPS of Baz/Par6 complex is required for their apical condensation during ACD of *Drosophila* larval NBs.** ToPro-3 in white. Red arrowheads indicate apical cortex. Scale bars, 5 μm. **a**, **b** Expressing *Flag-Baz WT* or mutant variants (**a**, using an actin "Flip-out" system marked by GFP) or *Flag-Par6 WT* or *ΔPB1* (**b**, using a UAS/GAL4 system driven by *insc-gal4*) in WT NBs of larval brains. Flag-Baz WT, Flag-Baz ΔPDZ2, and Flag-Par6 WT are localized on the apical cortex. Flag-Baz NTDmu and Flag-Baz PDZ3mu are diffused on the whole cortex and cytoplasm, whereas Flag-Par6 ΔPB1 is largely diffused in the cytoplasm. **c** Statistical data for **a**. For Flag-Baz WT, Flag-Baz NTDmu, Flag-Baz ΔPDZ2, and Flag-Baz PDZ3mu, $n = 11, 23, 19$, or 17 NBs collected from 30 larval brains for each genotype over three independent experiments, respectively. **d** Statistical data for **b**. For Flag-Par6 WT and Flag-Par6 ΔPB1, $n = 20$ or 19 NBs collected from ten larval brains for each genotype over three independent experiments, respectively. **e** Representative images showing Flag and Baz localization in NBs (marked by GFP using MARCM technique, see Methods section) from WT, *baz* mutant, or *baz* mutant expressing a *Flag-Baz WT* or *mutant* variants. **f** Statistical data for **e**. For WT, *baz*;Flag-Baz WT, *baz*;Flag-Baz NTDmu, *baz*;Flag-Baz ΔPDZ2, and *baz*;Flag-Baz PDZ3mu, $n = 20, 5, 5, 5$, and 8 NBs collected from 30 larval brains for each genotype over three independent experiments, respectively. **g** Representative images showing Flag and Par6 localization in NBs derived from WT, *par6* mutant, or *par6* mutant expressing a *Flag-Par6 WT* or *ΔPB1* mutant. **h** Statistical data for **g**. For WT, *par6*;Flag-Par6 WT and *par6*;Flag-Par6 ΔPB1, $n = 20, 5$, and 9 NBs collected from 30 larval brains for each genotype over three independent experiments, respectively. For all the statistical data, mean ± 95% confidence interval is shown; ns, not significant; *$p < 0.05$, **$p < 0.01$, ***$p < 0.001$, and ****$p < 0.0001$ using one-way ANOVA with Tukey's multiple comparison test. All the constructs are listed in Supplementary Table 1. Source data are provided as a Source data file.

*Flag-Baz PDZ3mu* rescued the NB lineage defects to a lesser extent (Fig. 8a, b). As a negative control, ectopic expression of *Flag-Par6* in *baz* NBs had no influence on the lineage defects. Similarly, a smaller lineage was generated in *par6* mutant NBs, while restoring

*Flag-Par6 WT* but not *Flag-Par6 ΔPB1* expression in *par6* mutant NBs could significantly rescue the defects (Fig. 8c, d). Ectopic expression of *Flag-Baz* was also included as a negative control.

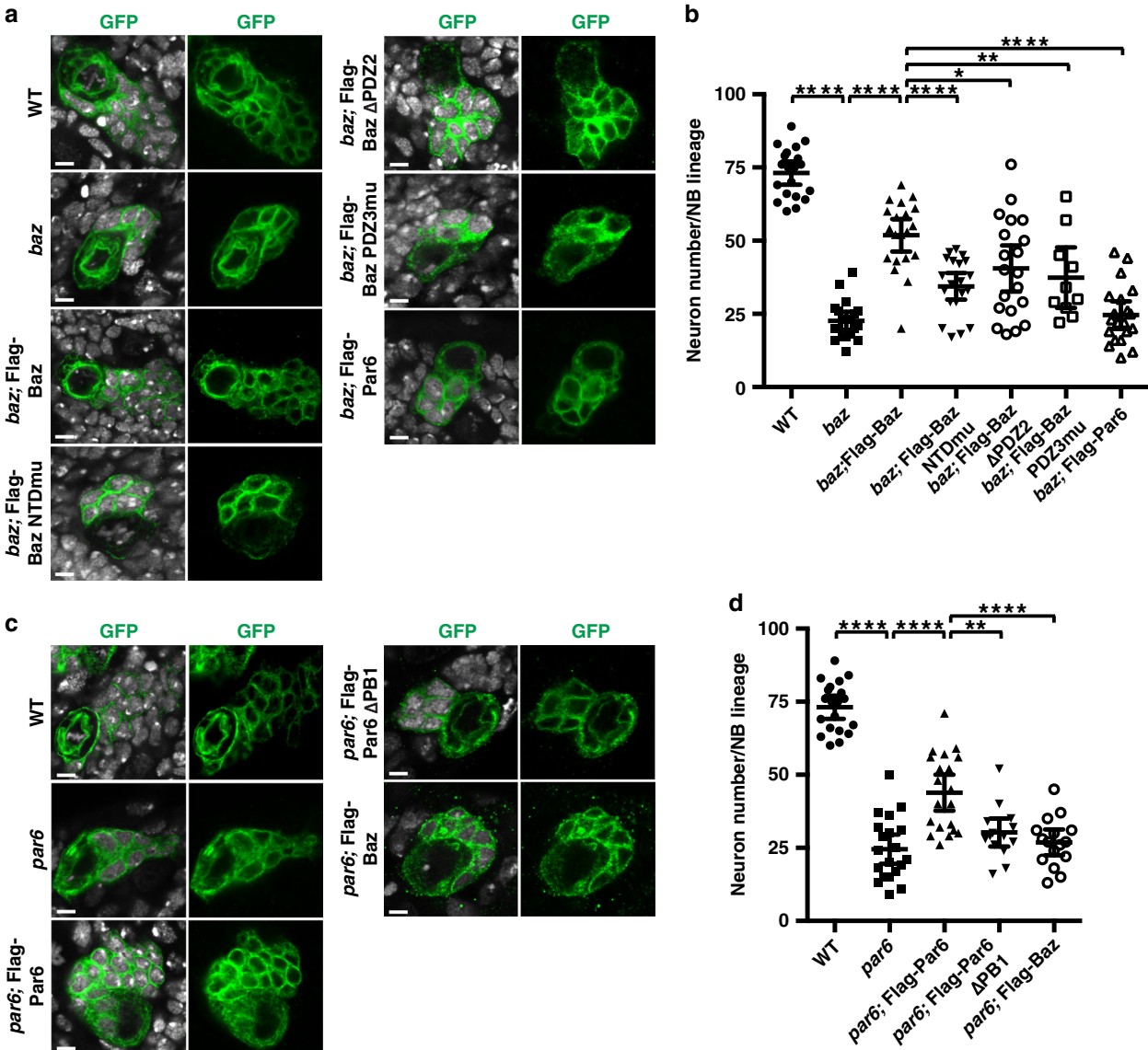

**Fig. 8 LLPS of Baz/Par6 is critical for neuronal differentiation in *Drosophila* NBs.** ToPro-3 is in white, GFP in green. Scale bars, 5 μm. **a** Representative images showing NB lineage marked by MARCM method for WT, *baz* mutant clone, or *baz* mutant clone rescued with WT or different variants. It is noted that the defective lineage development phenotype of *baz* mutant NBs could be largely rescued with *Flag-Baz WT* and *Flag-Baz ΔPDZ2*, and only partially rescued by the LLPS less efficient *Flag-Baz NTDmu* and *Flag-Baz PDZ3mu*, but could not be rescued with *Flag-Par6*. **b** Statistical data for **a**. For WT, *baz*, *baz*; Flag-Baz, *baz*;Flag-Baz NTDmu, *baz*;Flag-Baz ΔPDZ2, *baz*;Flag-Baz PDZ3mu, and *baz*;Flag-Par6, n = 20, 20, 20, 20, 20, 10, and 20 NBs collected from 30 larval brains for each genotype over three independent experiments, respectively. **c** Representative images showing that *par6* mutant NB lineage harboring less progeny that is largely reverted by expression of *Flag-Par6 WT* but cannot be rescued by the LLPS less efficient *Flag-Par6 ΔPB1* variant or *Flag-Baz*. **d** Statistical data for **c**. For WT, *par6*, *par6*;Flag-Par6, *par6*;Flag-Par6 ΔPB1, and *par6*;Flag-Baz, n = 20, 20, 20, 15, and 16 NBs collected from 30 larval brains for each genotype over three independent experiments, respectively. For all the statistical data, mean ± 95% confidence interval is shown. ns, not significant; *p < 0.05, **p < 0.01, ***p < 0.001, and ****p < 0.0001 using one-way ANOVA with Tukey's multiple comparison test. All the constructs are listed in Supplementary Table 1. Source data are provided as a Source data file.

The rescuing efficiency of Baz or Par6 apical localization and NB defective lineage phenotype in *baz* or *par6* mutant NBs by various Flag-Baz or Par6 variants was nicely correlated with their abilities to induce LLPS in heterologous cells and in vitro, demonstrating that the multivalent Baz-Par6 interaction-induced LLPS may be essential for the efficient Baz and Par6 enrichment in the apical crescent, and the correct ACD process in NBs.

**The nature of the Par complex clustering is LLPS.** Above biochemical, cellular, and genetic data had established a connection between in vitro Par proteins LLPS and in vivo Par complex local condensation; however, one thing hard to ignore is that, both the puncta formation assay in COS7 cells and the rescue assay in fly NBs were overexpression systems (Figs. 2 and 7). As Par3/Par6β LLPS was concentration dependent, though we have showed that Par3N/Par6β undergo LLPS at micromolar concentration (Fig. 3), we don't know the actual protein levels in dividing NBs. Through a fluorescent microscope imaging-based measurement method[49], we measured the average protein concentration of Par3N (when co-expressed with Par6β) in COS7 cells to be ~5 μM (vs ~40 μM in puncta; Supplementary Fig. 8a), which was high enough to

induce Par3 LLPS in the cytoplasm (Fig. 2). To measure the endogenous concentration of Baz protein, we used *GFP::baz* protein trap line that carries a GFP reporter gene inserted in the intronic region of *baz* locus and has been widely used in assaying Baz function[50]. The estimated average Baz level in fly NBs was only 0.14 μM (vs 1.32 μM in the crescent; Supplementary Fig. 8b), consistent with the fact that Baz did not undergo LLPS in the cytoplasm. Compared with the cytoplasmic fraction, we assumed that spatiotemporal attachment of the Par complex to the membrane led to its enrichment and subsequent LLPS in the apical domain of dividing NBs.

The NTD domain is known for Baz self-association to promote Par complex clustering. Thus, to further address functional significance of LLPS of Baz at physiological level without interfering other protein–protein or protein–membrane interactions, we generated GFP knock-in *Drosophila* strains expressing endogenous levels of GFP-tagged Baz ΔNTD, FUSs-Baz, or FUS$_L$-Baz chimera mutant (Baz NTD was replaced by FUS$_S$ or FUS$_L$, respectively; Supplementary Table 1). In sharp contrast to the early UAS/GAL4-based rescue assay results showing that Baz NTDmu lost its apical condensation in both WT and *baz* mutant backgrounds (Fig. 7), the knock-in mutant Baz ΔNTD showed obvious apical condensation in all NBs (Fig. 9a, b). We assumed that the distinct phenotypes of these variants arose from the different levels of protein expressed in these different assays (overexpression for Baz NTDmu vs endogenous level for Baz ΔNTD, see Discussion section for more details). However, we noticed that the asymmetrically localized proteins, including apical Baz ΔNTD, Par6 and aPKC, and basal Mira showed less condensed localization and significant cytoplasmic diffusion in *Baz ΔNTD* knock-in NBs compared to *Baz WT* knock-in NBs (Fig. 9a, b), implying that Baz LLPS-mediated by NTD (note that NTD oligomerization is only one driving force for Baz LLPS) at physiological condition indeed facilitates the efficient Baz apical localization and thus regulated local condensation of other proteins. In line with these defects, although *Baz ΔNTD* knock-in was viable and able to establish a homozygous stock, the L3 larval brain was however significantly smaller than that of *Baz WT* larvae (Fig. 9c). For some unidentified reason, *FUS$_L$-Baz* knock-in strain was lethal and excluded from this study. *FUSs-Baz* knock-in strain was viable. Though the LLPS less efficient FUS$_S$-Par3N only partially restored the puncta formation property of Par3N when co-expressed with Par6β in COS7 cells (Fig. 5e–g), FUSs-Baz exhibited apical localization in dividing NBs, similar to that of *Baz WT* knock-in (Fig. 9a, b). Importantly, the *FUSs-Baz* NBs showed largely normal localization of both apical (Par6 and aPKC) and basal (Mira) proteins. L3 larval brain of *FUSs-Baz* larvae was also significantly larger than that of *Baz ΔNTD* knock-in (Fig. 9c, d). In addition, we confirmed that the *FUSs-Baz* mutant had a comparable FRAP recovery rate as Baz WT (Fig. 9e–g). Collectively, the above data strongly demonstrated that NTD-mediated Par clustering in dividing NBs relied on its oligomerization-dependent LLPS behavior.

## Discussion

How the conserved Par (Par3/Par6/aPKC) complex is selectively recruited and concentrated on membranes for polarity establishment remains unclear. In this study, different from previously reported crescent localization patterning[25–27], we reveal that the endogenous Par complex exhibits cell cycle-dependent discrete puncta formation on the apical cortex in *Drosophila* NBs. The condensed Par puncta emerge from prophase, further condensate and enlarge as a clustered puncta structure in metaphase, then subsequently disassemble into scattered small puncta from anaphase (Fig. 9h). The cell cycle-dependent clustering of Par

proteins in *Drosophila* NBs were also observed by two recent studies[51,52]. Our in vitro biochemical data together with heterologous cell-based studies showed that the Par3/Par6 complex can undergo LLPS at very low protein concentrations (Fig. 3a, d), and mutations of Par3 or Par6 that impair LLPS were found to alter ACD in *Drosophila* NBs. It has been recently shown that the basal condensation of Numb in dividing NBs is also regulated by LLPS of the Numb/Pon complex[53]. Thus, LLPS may be a common mechanism for the local condensation of apical and basal polarity determining protein complexes.

It is important to note that the Par proteins, each at their endogenous level, can form clustered puncta via LLPS on the cortex (Fig. 1). Though the measured endogenous Baz level in *Drosophila* NBs was too low to induce its LLPS in the cytoplasm, two-dimensional membrane attachment was expected to locally enrich the protein and lead to its LLPS (Supplementary Fig. 8b). In return, LLPS-mediated Par complex condensates formation acts as an effective way for cells to further concentrate limited amount of Par proteins to specific cell cortices for polarity establishment. We propose that apical Baz/Par3 localization is a balanced result of apical anchoring and LLPS-mediated local condensation (via multivalent protein–protein interaction, self-association, protein–membrane interaction, etc.). Thus, for the knock-in mutant Baz ΔNTD, partially impaired LLPS ability due to its defective oligomerization led to its less condensed localization and significant cytoplasmic diffusion. However, the situation was different for the overexpressed Baz NTDmu (driven by UAS/GAL4) in the rescue assay, which is ectopically localized. As LLPS is very sensitive to concentrations of biological components, an overexpression of Par proteins especially Baz/Par3, the core driving factor of LLPS, may cause artificial promotion of the Par complex condensation via LLPS. Whereas the apical anchoring capacity of NBs seems to have a limitation. In UAS/GAL4-based rescue assay, the overexpressed Baz WT phase condensates may just have reached the threshold of apical anchoring capacity, whereas the LLPS deficient, overexpressed Baz NTDmu broke the balance, and led to its cortical and cytoplasmic diffusion. If the expression level goes higher, even Baz WT can not be afforded apically. Consistent with this notion, high *Flag-Baz* expression in a WT background (driven by *insc-gal4*), has a dominant-negative effect and leads to ectopic localization of endogenous Par complex throughout the cortex, and consequently disrupts localization of basal proteins (Supplementary Fig. 7c, d). Similarly, ectopic Baz localization was observed when exogenous Baz is forcedly expressed in embryonic NBs (refs. [25,54]). It was recently shown that overexpression of Par3-induced cell polarity in apolar S2 cells by forming concentrated Par-dots that further fused into amorphous Par-islands[51]. According to a study of protein LLPS on lipid membrane bilayers, protein clusters gradually grew and fused into larger ones with irregular shapes, and finally coalesced into a mesh-like network[49]. Thus, the amorphous structure of Par-islands in S2 cells may arise from the overexpression and overaccumulation of Par3 in the membrane region. Therefore, caution should be taken in interpreting the overexpression phenotypes of Par3.

Another key finding in this study is that aPKC can be recruited and concentrated in Par3/Par6 condensates as an inactive client. Such condensed phase droplets could be an efficient mechanism for local condensation of aPKC (ref. [31]). Spatiotemporal activation of aPKC (e.g., by Cdc42) and consequent phosphorylation on Par3 CR3 leads to disassembly of the Par complex condensates (Fig. 9h). Another cell cycle regulator that might play a role in Par LLPS regulation is Plk1, which inhibits the oligomerization of Par3 by phosphorylating NTD in *C. elegans*[32]. A critical but currently unknown point is how the autoinhibition of Par3 is relieved, as the open conformation of Par3 is critical for the Par

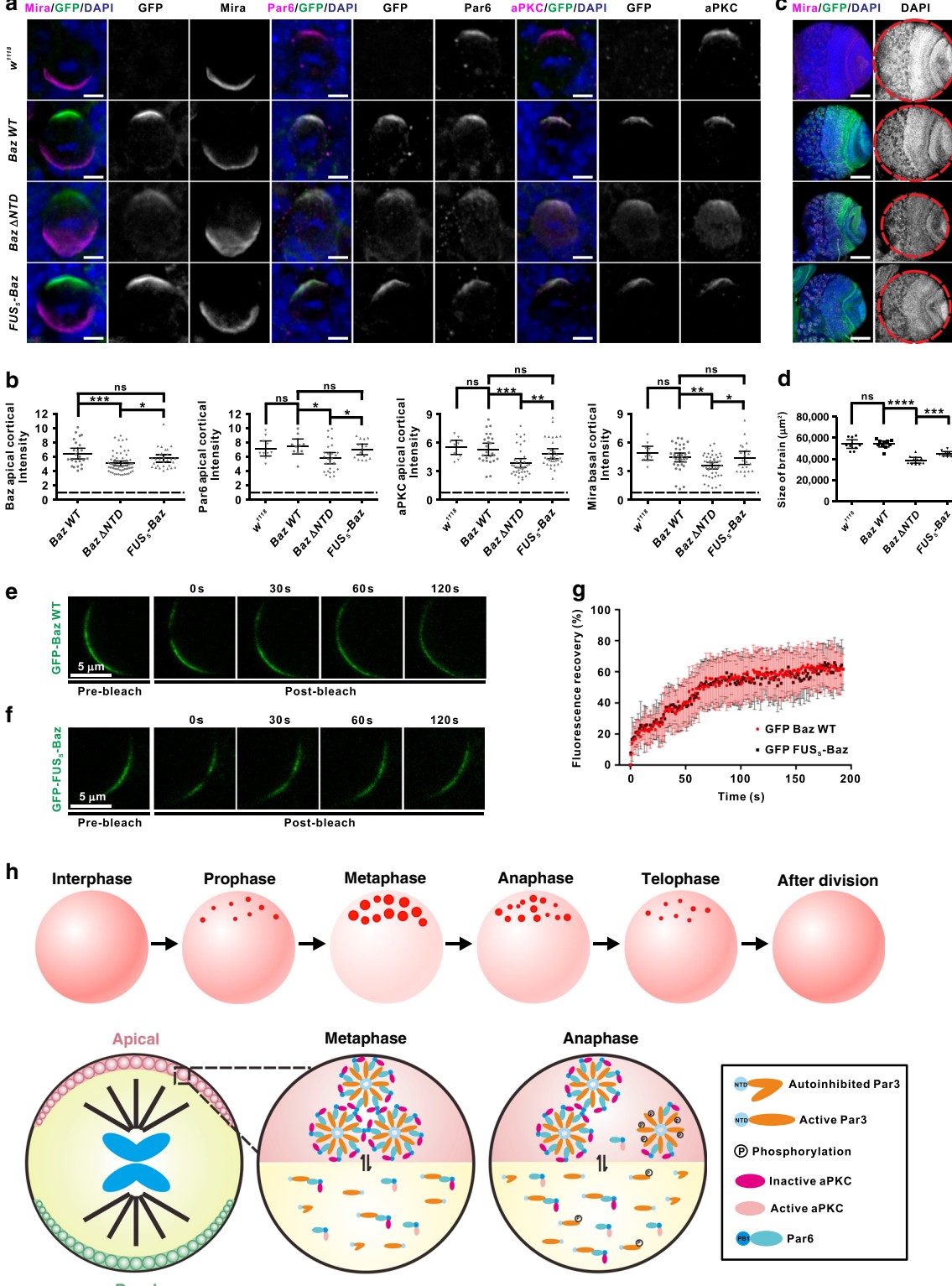

complex condensate formation. Nonetheless, it is likely that multilayered regulatory mechanisms can act concertedly to control the spatiotemporal assembly and disassembly of the Par complex phase separation, and hence the cell polarity regulations.

It is increasingly recognized that LLPS is a common strategy for cells to form membrane-less compartments by selectively recruiting and condensing proteins/RNAs/lipids[55–59]. In a broader sense, the local condensation of other master polarity complexes, such as the conserved Lgl/Dlg/Scribble complex in the apical–basal polarity, and the Prickle/Vangl and Frizzled/Disheveled/Diego complexes in the planar cell polarity, may adopt a similar LLPS-driven mechanism to establish cell polarity in different tissues. Like the Par complex proteins, all these complexes share several common features: (1) these proteins contain multiple domains, which mutually interact with each other or self-associate in vitro to form complex platform, which further recruits other binding partners to

**Fig. 9 NTD-mediated Par clustering relies on its oligomerization-dependent LLPS behavior. a** Representative images of knock-in larval NBs expressing GFP-tagged Baz WT, Baz ΔNTD, or FUS$_S$-Baz chimera visualized with GFP, Par6, aPKC, or Mira. ToPro-3 in blue. Scale bars, 5 μm. **b** Statistical analysis of ACD protein localization for **a**. For Baz apical cortical intensity, Baz WT, Baz ΔNTD, and FUS$_S$-Baz, $n = 24$, 67, and 29 NBs collected from 30 larval brains for each genotype over three independent experiments, respectively. For Par6 apical cortical intensity, $w^{1118}$, Baz WT, Baz ΔNTD, and FUS$_S$-Baz, $n = 10$, 10, 26, and 14 NBs collected from 15 larval brains for each genotype over three independent experiments, respectively. For aPKC apical cortical intensity, $w^{1118}$, Baz WT, Baz ΔNTD, and FUS$_S$-Baz, $n = 10$, 25, 39, and 32 NBs collected from 15 larval brains for each genotype over three independent experiments, respectively. For Mira basal cortical intensity, $w^{1118}$, Baz WT, Baz ΔNTD, and FUS$_S$-Baz, $n = 10$, 34, 46, and 21 NBs collected from 15 larval brains for each genotype over three independent experiments, respectively. **c** Representative images showing overview of larval brains expressing knock-in GFP-tagged Baz WT, Baz ΔNTD, or FUS$_S$-Baz chimera. **d** Statistical data measuring brain size presented in for **c**. $n = 10$ larval brains over three independent experiments. **e, f** Representative time-lapse FRAP images showing that recovery of knock-in GFP-Baz WT **e** or GFP-FUS$_S$-Baz **f** signal within the preformed crescent occurred within a few minutes. **g** Statistical data for **e** and **f**. For GFP-Baz WT and GFP-FUS$_S$-Baz, $n = 13$ and 9 NBs collected from 15 larval brains for each genotype over three independent experiments, respectively. **h** Model for Par proteins local condensation during the ACD of *Drosophila* NBs. For simplicity, the basal daughter cell was omitted. All the constructs are listed in Supplementary Table 1. For all the statistical data, mean ± 95% confidence interval is shown. ns, not significant; *$p < 0.05$, **$p < 0.01$, ***$p < 0.001$, and ****$p < 0.0001$ using one-way ANOVA with Tukey's multiple comparison test. Source data are provided as a Source data file.

assemble into higher order protein interaction network; (2) these complexes are found to form condensed patches or puncta attached to the inner surface of plasma membranes in vivo[60]; and (3) proteins within these condensed patches or puncta are highly dynamic and rapidly exchange with corresponding proteins in the cytoplasm. The multivalent interaction-induced LLPS theory can perfectly explain above phenomena[61,62], allowing the stable existence of large concentration gradients of the proteins within the local protein condensates and those in the cytoplasm, and at the same time, keeping the proteins in the condensed phase highly dynamic. Such dynamic association may be essential for the fast assembly/dis-assembly of these polarity complexes in responding to extrinsic/intrinsic cues/signals to rearrange the cell polarity. We postulate that LLPS of polarity protein complexes induced by multivalent interactions is a general mechanism for the cell polarization.

## Methods

**Protein expression and purification.** Various rat Par3 fragments (Uniprot ID: Q9Z340, Supplementary Tables 1 and 2), mouse Par6β fragments (Uniprot ID: Q9JK83), and the mouse PKCι PB1 (Uniprot ID: Q62074, aa 16–99) were individually cloned into pGEX-6P-1 or a modified version of pET32a vector[53]. All the mutations used in this study were generated using the standard PCR-based mutagenesis method and confirmed by DNA sequencing. Human FUS LCD (Uniprot ID: P35637, aa 1–214 or 1–141) or p62 PB1 (Uniprot ID: Q13501, aa 1–102) was fused to Par3N ΔNTD using the standard PCR-based mutagenesis method and confirmed by DNA sequencing. Recombinant proteins were expressed in *Escherichia coli* BL21 (DE3) or Rosetta host cells in LB medium at 16 °C, and purified using a Ni$^{2+}$-NTA agarose affinity column followed by SEC (using HiLoad 26/600 superdex 75/200 pg columns on an AKTA Fast Protein Liquid Chromatography (FPLC) system, GE Healthcare) with buffer A containing 50 mM Tris (pH 8.0), 100 mM NaCl (300 mM NaCl for Par3N), 1 mM EDTA, and 1 mM dithiothreitol (DTT). Par6α PBM peptide (MRGDVSGFSL), Par6β PBM peptide (LEEDGTIITL), and Par6γ PBM peptide (VEEHGPAITL) were commercially synthesized (Phtdpeptides).

**GST pull-down assay.** GST or GST fusion proteins (4 nmol) was first loaded onto 30 μl GSH-Sepharose 4B slurry beads and then incubated with 12 nmol indicated proteins in 500 μl buffer A at 4 °C for 1 h. After being washed three times with the same buffer, the above proteins captured by affinity beads were eluted by boiling with SDS-loading buffer, resolved by 12% SDS–polyacrylamide gel electrophoresis (SDS–PAGE), and detected by Coomassie blue staining.

**Cell lysate GST pull-down assay and immunoblotting.** Human HEK293T cells (from ATCC) were cultured in Dulbecco's modified Eagle medium (Hyclone) containing 10% fetal bovine serum (FBS, Gibco). Cells were transiently transfected with 6 μg of Flag-tagged Par3N or Par6β using polyethylenimine transfection reagent (Polysciences). Cells were harvested 36 h post transfection and lysed in the buffer containing 50 mM Tris (pH 7.4), 150 mM sodium chloride, 1% Nonidet P-40, and protease inhibitor cocktail (APExBIO). Each lysate was incubated with GST fusion proteins at 4 °C for 2 h.

After extensive washing of the beads with the lysis buffer, bound proteins were boiled in SDS–PAGE loading buffer and subjected to SDS–PAGE. Proteins were transferred to a 0.45 μm polyvinylidene difluoride (PVDF) membrane (Millipore), which was blocked using 3% bovine serum albumin in TBST (20 mM Tris-HCl (pH 7.4), 137 mM NaCl, and 0.1% Tween-20) buffer at room temperature for 1 h,

followed by incubation with the anti-Flag (ABclonal, AE005) at a 1/2000 dilution at 4 °C overnight. Membrane were washed three times with TBST buffer, incubated with horseradish peroxidase-conjugated goat anti-mouse antibody (ABclonal, AS003) or anti-rabbit antibody (ABclonal, AS014), at a 1/5000 dilution, at room temperature for 1 h, and visualized on a LAS4000 chemiluminescent imaging system.

**Fluorescence polarization assay.** Fluorescence polarization assays were performed on a PerkinElmer LS-55 fluorimeter equipped with an automated polarizer at 25 °C. In a typical assay, a FITC (Molecular Probes)-labeled peptide (~1 μM) was titrated with a binding partner (Par3 PDZs) in buffer A. For the 1,6-hexanediol reversing experiment, Par3 PDZ3 was preincubated with indicated amounts of 1,6-hexanediol.

**Crystallography.** Freshly purified rat Par3 PDZ3 was concentrated to 7–15 mg/ml before adding Par6β PBM peptide (1 mM stock solution in buffer A) with a molar ratio of 1:3. Crystals of the Par3 PDZ3/Par6β PBM complex were grown by the hanging drop vapor diffusion method at 16 °C in a reservoir solution containing 2.4 M ammonium sulfate and 0.1 M citrate (pH 4.0). The crystals were soaked in crystallization solution containing 20% glycerol for cryoprotection. The diffraction data of the crystals was collected at the beamline BL17U1 at Shanghai Synchrotron Radiation Facility in China (SSRF) at wavelength of 0.9792 Å. The data were processed and scaled using HKL2000 (ref. [63]). The phase problem of the Par3 PDZ3/Par6β PBM complex was solved by molecular replacement using the program PHASER[64] with the solution structure of Par3 PDZ3 (PDB ID: 2K1Z) as the search model before was adjusted by COOT[65]. The initial model was further rebuilt, adjusted manually with COOT, and refined by the phenix.refine program of PHENIX[66]. The statistics of the data collection and final refinement statistics are summarized in Table 1. The PDB accession code is 6JUE.

**Analytical gel filtration chromatography.** Analytical gel filtration chromatography was carried out on an AKTA FPLC system (GE Healthcare). Protein samples were pre-concentrated to indicated concentrations and then loaded on a Superdex$^{TM}$ 200 increase 10/300 GL column (GE Healthcare) pre-equilibrated with the buffer containing 50 mM Tris-HCl (pH 8.0), 100 mM (or indicated concentrations) NaCl, 1 mM DTT, and 1 mM EDTA.

**In vitro phase transition assay.** Various Par3, Par6β, and PKCι fragments were prepared in buffer A precleared via high-speed centrifugations. In this assay, the proteins were mixed at indicated molar radio at final concentrations spanning 0.5–25 μM. Formations of phase transition were assayed either directly by imaging-based methods or by sedimentation-based methods.

For imaging, mixtures were observed by being injected into a homemade flow chamber comprised of a glass slide sandwiched by a coverslip with one layer of double-sided tape as a spacer for DIC or fluorescent imaging (Leica TCS SP5). For the sedimentation assay, samples were subjected to centrifugation at $21,130 \times g$ for 10 min. Supernatant was isolated from pellet into a clean tube immediately after centrifugation. The pellet fraction was washed once with buffer A and thoroughly resuspended with the same buffer to the equal volume as supernatant fraction. Proteins from both fractions were detected by 12% SDS–PAGE with Coomassie blue staining. Band intensities were quantified using the ImageJ software.

For fluorescence assay, Par3N, Par6β, and PKCι PB1 were purified in buffer containing 100 mM NaHCO$_3$ (pH 8.3), 100 mM NaCl (300 mM NaCl for Par3N), 1 mM EDTA, and 1 mM DTT. iFluor$^{TM}$ 488 NHS ester, Cy3 NHS ester, or 405 NHS ester (AAT Bioquest) were incubated with Par3N, Par6β, or PKCι PB1, respectively, at room temperature for 1 h (fluorophore to protein molar ratio was 1:1). Reaction was quenched by 200 mM Tris (pH 8.0). Chemical-labeled proteins were further purified into buffer A by Hitrap desalting column. In flow chamber at room temperature, mixture of Par3N (25 μM, with iFluor$^{TM}$ 488-labeled Par3N

**Table 1 Data collection and refinement statistics.**

|  | Par3 PDZ3/Par6b PBM peptide |
|---|---|
| **Data collection** | |
| Space group | $P6_3$ |
| Cell dimensions | |
| $a, b, c$ (Å) | 57.454, 57.454, 52.313 |
| $\alpha, \beta, \gamma$ (°) | 90.000, 90.000, 120.000 |
| Wavelength (Å) | 0.9792 |
| Resolution (Å) | 49.76–1.55 (1.63–1.55)[a] |
| $R_{merge}$ (%) | 8.9 (50.3) |
| Mean $I/\sigma$ | 14.8 (3.0) |
| Completeness (%) | 99.5 (98.0) |
| Redundancy | 9.0 (4.8) |
| **Refinement** | |
| Resolution (Å) | 49.76–1.55 |
| No. reflections | 14250 |
| $R_{work}/R_{free}$ (%) | 21.09/22.26 |
| No. atoms | |
| Protein | 731 |
| Water | 29 |
| B factors | |
| Protein | 25.71 |
| Water | 26.20 |
| R.m.s deviations | |
| Bond lengths (Å) | 0.005 |
| Bond angles (°) | 0.729 |

[a]Values in parenthese indicate the highest-resolution shell.

mixed with 60 molar ratios of unlabeled molecules) and Par6β (25 μM, with Cy3-labeled Par6β mixed with 300 molar ratios of unlabeled molecules) or PKCι PB1 (25 μM, with 405-labeled PKCι PB1 mixed with 100 molar ratios of unlabeled molecules) was observed with a Leica TCS SP5 confocal microscope.

**Turbidity assay**. Various protein concentrations of FUS LCD (1–214/1–141) were prepared in buffer A adding 10% PEG8000 in a 96-well microplate. The absorption (turbidity) of samples were measured at 600 nm in a SpectraMax microplate reader. Results were recorded in quintuplicate for each protein sample. All assays were performed in quintuplicate ($n = 5$).

**COS7 cell imaging and data analysis**. For each well in a six-well plate, various Par3, Par6, and PKCι plasmids were individually or co-transfected into COS7 cells (from ATCC) using polyethylenimine transfection reagent. Cells were fixed by 4% paraformaldehyde and mounted on glass slides for imaging using a Leica TCS SP5 confocal microscope by a 64X oil-immersion lens with DAPI staining. Confocal images were processed with ImageJ. For puncta-counting assay, data were collected from four to six independent batches of cultures as indicated in the figures. In each batch, at least 600 fluorescence-positive cells were counted for each group of experiments. A cell with more than two obvious fluorescence puncta was counted as a puncta-positive cell. Experiments were conducted in a blinded fashion.

**FRAP assay**. The in vitro FRAP analysis of iFluor488-Par3N droplets was carried out in a 1:1 mixture of Par3N and Par6β (25 μM) at room temperature. The 488 signal was bleached using a 488-nm laser beam with a Leica TCS SP5 confocal microscope.

COS7 cells were cultured in glass bottom dishes and transfected as described above. FRAP assay was performed on a Leica TCS SP5 confocal microscope. Puncta with diameters ~1.0 μm were assayed. GFP signal was bleached using a 488-nm laser beam. The fluorescence intensity difference between pre-bleaching and at time 0 (the time point right after photobleaching pulse) was normalized to 100%. The experimental control is to quantify fluorescence intensities of similar puncta/cytoplasm regions without photobleaching.

**In vitro kinase assay**. HEK293T cells transfected with the Flag-tagged PKCι WT or A120E was lysed. After centrifugation, the lysate was mixed with anti-Flag M2 affinity gel (Sigma) at 4 °C for 3 h. The anti-Flag M2 affinity gel-captured PKCι was washed twice with the lysis buffer, once with the kinase assay buffer (50 mM Tris, 100 mM NaCl, 10 mM MgCl2, 150 μM ATP, pH 8.0), and then released by adding the Flag peptide to a final concentration of 0.1 mg/ml. In each kinase reaction assay, Par3 1–854 (3 μM) and Par6β (3 μM) were mixed with PKCι in 100 μl kinase assay buffer at room temperature for 5 min before centrifugation at $21,130 \times g$ for 10 min. Supernatant was isolated from pellet into a clean tube immediately after

centrifugation. The pellet fraction was washed once with kinase assay buffer and thoroughly resuspended with the same buffer to the equal volume as supernatant fraction. Proteins from both fractions were detected by Phos-tag SDS–PAGE with Coomassie blue staining. Phos-tag SDS–PAGE was performed by using a resolving gel containing 50 μM phos-tag acrylamide (NARD, AAL-107) and 0.1 mM MnCl2. The running buffer consisted of 25 mM Tris, 192 mM glycine, and 0.1% (w/v) SDS.

**Quantification of protein concentrations in cells**. To generate a standard calibration curve for directly converting the measured florescence intensity into absolute protein concentration, purified GFP at various indicated concentrations were injected into the homemade flow chamber, and the fluorescence intensity at each concentration was measured using the same imaging parameters (Olympus IX73 microscope, 12-bit image and exposure 60 ms for COS7 cells; Zeiss 780 upright confocal microscope, 16-bit image, laser power 2.0, and gain value 850 for NBs), as for the whole cell or puncta/crescent quantifications. With these calibration curves, the measured fluorescence intensity for GFP-Par3N in COS7 cells or GFP-Baz in NBs can be converted into absolute molar concentration, according to each standard calibration curve. Identical parameters (laser power, detector gain, bit depth, and exposure time) were used during the imaging processes for COS7 cells or NBs, respectively. Images were analyzed by the ImageJ software.

***Drosophila* S2 cell culture**. Various Baz (WT gene is a gift from Fumio Matsuzaki and Daniel St Johnston) and Par6 (WT gene is a gift from Fumio Matsuzaki) fragments were subcloned into UASt.attB vector (a gift from Konrad Basler).

*Drosophila* S2 cells (from *Drosophila* Genomics Resource Center) were cultured with Shields and Sang M3 insect medium (Sigma) supplemented with 10% FBS at 25 °C. Effectene Reagent (Qiagen) was used in all transfections following the manufacturer's instructions. In short, the ratio of DNA to effectene was sustained at 1:20. The S2 cells were co-transfected with 0.5 μg plasmids of interest (Baz or Par6 variants) together with *act-gal4* plasmid. After 48 h post transfection, S2 cells were harvested and lysed in Nonidet P-40 lysis buffer containing 50 mM Tris (pH 8.0), 250 mM NaCl, 0.5% Nonidet P-40, 0.2 mM EDTA, protease inhibitor cocktail (complete, Roche), and phosphatase inhibitor. The cell lysate was spun by microcentrifugation at maximum speed for 5 min at 4 °C. The samples were separated by 10% polyacrylamide SDS–PAGE gels followed by transferring to PVDF membrane (Millipore). Mouse anti-Flag antibody (Sigma, F1804, 1/2000), rabbit anti-Baz antibody (generated in our lab, 1/1000), and rabbit anti-Par6 (generated in our lab, 1/2000) were diluted in TBST with 5% non-fat dry milk.

**Fly genetics**. The fly stocks and crosses were maintained at 25 °C on standard medium. Detailed information of the fly stocks used in this study was provided in the following description or FlyBase (www.flybase.org). Fly stocks were attained from Blooming Drosophila Stock Center, unless otherwise stated below.

Stocks used in the study included FRT19A, elav-gal4, insc-gal4, Ay-gal4 (*actin flip-out*), Tub-gal80, hs-flp, UAS-CD8::GFP, baz[4] (a gift from Andreas Wodarz), and par6[Δ226] (a gift from Juergen A Knoblich), GFP::baz[50].

Baz and Par6 variants were cloned in UASt.attB vector and transgenic stocks with specific insertion site were obtained through target insertion utilizing attP landing site on III chromosome (Best Gene line 9732) or II chromosome (Best Gene strain 9723), respectively, by BestGene, Inc. (ChinoHills, CA).

Mosaic analysis with a repressible cell marker (MARCM) technique was used to positively mark mutant clones with a GFP signal according to published protocol[67]. In brief, embryos were collected over a time window of 6 h, and larvae (24 h after larval hatching, ALH) was exposed to heat shock treatment for 1 h at 37 °C, and larvae with desired genotypes were dissected and examined.

**Immunohistochemistry and imaging**. Larvae of desired genotype were dissected at 96 h ALH and brains were fixed for 15 min in 3.7% formaldehyde in PBS with 0.1% Triton-X, and subsequently processed for immunochemistry analysis. The following antibodies were used: mouse anti-Flag (Sigma, F1804, 1/2000; rabbit anti-Mira[68] (generated in our lab), 1/1000; guinea-pig anti-Dpn (ref. [68]; generated in our lab), 1/1000; rabbit anti-Baz (generated in our lab), 1/2000; rabbit anti-Par6 (generated in our lab), 1/1000; rabbit anti-aPKCζ C20 (Santa Cruz Biotechnologies, SG-216-G), 1/1000; chicken anti-GFP (Abcam, ab13970), 1/5000; rabbit anti-Pon (generated in our lab), 1/2000; and rabbit anti-Numb (a gift from Xiaohang Yang), 1/1000. Secondary antibodies were conjugated with Alexa Fluor 488, Alexa Fluor 555, and Alexa Fluor 633 (Molecular Probes) used at 1/500, 1/1000, and 1/250, respectively. TO-PRO-3 (Invitrogen) or Hoechst 33342 (Sigma) was used at 1/5000 for DNA staining, and samples were mounted in Vectashield (Vector Laboratories). Images were obtained using Leica SP8 confocal microscope or Zeiss LSM 780 laser scanning confocal microscope and processed in Adobe Photoshop CC2018.

**Effect of 1,6-hexanediol on Par assemblies**. To analyze the effect of 1,6-hexanediol on Baz/Par6 assemblies in vivo, larvae of $w^{1118}$ at 96 h ALH were dissected in ice cold Shields and Sang M3 insect medium (Sigma), and larvae brains were transferred to Shields and Sang M3 insect medium containing 0, 5, or 10% 1,6-hexanediol, and incubated for 2 min. The treated brains were fixed immediately for 15 min in 3.7% formaldehyde in PBS with 0.1% Triton-X, and then processed for immunochemistry analysis. For recovery experiment, the 1,6-hexanediol-treated

brains were washed with M3 insect medium for several times, and incubated in Shields and Sang M3 insect medium for 20 min before fixation and staining. Baz and Par6 localization were analyzed under Leica Confocal microscopy SP8 system. Only mitotic NBs were analyzed.

**Image analysis of ASI.** The asymmetric index (ASI) for membrane-associated ACD proteins is defined by previous reference[31] using the following equation (1):

$$\frac{A - B}{2(A + B)}$$

Where $A$ and $B$ are apical and basal signal intensity measured by ImageJ, respectively. For each experiment, the raw ASI values are normalized with the mean ASI of control. In this case, a value of 1 stands for WT asymmetry, whereas 0 stands for complete loss of asymmetry.

**Image analysis of ACI.** In some cases, apical or basal proteins have cytoplasmic localization. Then the measurement of apical or basal protein cortical intensity (ACI) follows previous reference[31] and is calculated by the mean greyscale value of each protein on the apical or basal crescent normalized by the mean greyscale value of protein in the cytoplasmic region. A value of 1 means uniform distribution of the protein throughout cytoplasm and cortex, whereas a value >1 stands for its condensation on the membrane.

**CRISPR CAS9 fly assay.** CRISPR/CAS9 system was used to generate *baz* GFP knocked-in stocks according to published protocol[69]. In brief, *baz* GFP knocked-in insertions were designed so that GFP tags all different splicing isoforms. gRNA sequence was cloned into pCFD4 vector and GFP knocked in *baz* donor templates were cloned into pHD-DsRed-attP vector. The gRNA constructs and donor templates were co-injected into *vasa-Cas9* transgenic embryos (BDSC#51324). Injections were performed by Tsinghua Fly center. For GFP-Baz ΔNTD, nucleotides of *baz* at position 127–270 of the PA isoform was replaced with in frame GFP coding. For GFP-FUS$_L$-Baz and GFP-FUS$_S$-Baz, nucleotides of *baz* at position 127–270 of the PA isoform was replaced with in frame inserted GFP-linker-FUS$_L$ and GFP-linker-FUSs sequence, respectively. gRNA construct and donor template was generated using primers listed in Supplementary Table 2.

**Quantification and statistical analysis.** Statistical parameters including the definitions and exact values of $n$ (e.g., number of experiments, number of cells, number of droplets, etc.) are reported in the figures or corresponding figure legends. Data of COS7 cell culture were expressed as mean ± SEM; ns, not significant; $*p < 0.05$, $**p < 0.01$, $***p < 0.001$, and $****p < 0.0001$ using one-way analysis of variance (ANOVA) with Tukey's multiple comparison test. Data of in vitro phase transition sedimentation assay and FRAP assay were expressed as mean ± SD. Data are judged to be statistically significant when $p < 0.05$ by one-way ANOVA with Tukey's multiple comparison test. None of the data were removed from our statistical analysis as outliers. For protein localization in NBs of various genetic backgrounds, only mitotic NBs were imaged and counted. For counting of cell numbers in a MARCM lineage, $z$-series images were acquired and cells were counted. All statistical data were conducted in GraphPad Prism 6. All experiments related to cell cultures and imaging studies were performed in blinded fashion.

**Reporting summary.** Further information on research design is available in the Nature Research Reporting Summary linked to this article.

## Data availability

The authors declare that all data supporting the findings of this study are available within the article and its Supplementary Information files or from the corresponding author upon request. Coordinates of the crystal structure of Par3 PDZ3/Par6β PBM complex have been deposited in the Protein Data Bank (http://www.rcsb.org/) under the accession code 6JUE. FlyBase dataset is available online at http://flybase.org. Source data file is available.

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

## Acknowledgements

We thank the staff of beamlines BL17U1 and BL18U1 at SSRF and –National Center for Protein Sciences Shanghai for data collection; Jiayi Zhang and Yanjun Liu for helping with DIC and fluorescence imaging; and Fumio Matsuzaki, Daniel St Johnston, Konrad Basler, Andreas Wodarz, Juergen A Knoblich, Xiaohang Yang, the Bloomington Stock Center for antibodies, vector and stocks. This work was supported by grants from the Ministry of Science and Technology of China (2019YFA0508401) and the National Natural Science Foundation of China (31871394 and 31670730) to W.W., Shanghai Municipal Science and Technology Major Project (2018SHZDZX01) and ZJLab, the National Natural Science Foundation of China (31972893) and Municipal Natural Science Foundation of Beijing (KZ201910028040) to Z.Li, RGC of Hong Kong (T13-605/18W) to M.Z., and Temasek Life Sciences Laboratory and Singapore Millennium Foundation to Y.C.

## Author contributions

Z.Liu, Y.Y., A.G., Y.C., and W.W. conceived the research and analyzed data. Z.Liu, Y.Y., A.G., J.X., H.L., W.H., Q.-Y.L., and Z.Li performed experiments. Z.Liu, Y.Y., A.G., Y.C., and W.W. created figures. Z.Liu, Y.M., M.Z., Y.C., and W.W. wrote the manuscript. W.W. coordinated the research.

## Competing interests

The authors declare no competing interests.
