## [Peer Review File · Nature Communications]

Reviewers' comments:

Reviewer #1 (Remarks to the Author):

This paper reports the phase separation of Par complex components into a liquid-like state *in vitro*. Structural and protein interaction studies characterize contributing mechanisms. Studies also reveal that the proteins of interest can form large aggregates with continual turnover in cell culture, and these studies allowed further characterization of the molecular mechanisms contributing to the aggregation (including the oligomerization domain of Par-3, an interaction site between Par-3 and Par-6, and oligomerization of the Par-6 PB1 domain). Par-3 can form the aggregates on its own, but Par-6 is found to enhance their formation. aPKC also joins the aggregates and aPKC phosphorylation of Par-3 seems to reduce its ability to form aggregates. A chemical inhibitor of liquid phase separation disrupts the Par-3-Par-6 phase separation *in vitro*, and the formation of Par-3 puncta in dividing *Drosophila* neuroblasts. Further suggesting a role for the phase separation in the dividing neuroblasts, alterations to Par-3 or Par-6 that disrupt their phase separation also disrupt their polarization in neuroblasts, as well as the division of the neuroblasts.

Overall, this paper presents a compelling case for Par complex phase separation important for asymmetric cell division. This finding would be of substantial interest. However, a number of issues should be addressed.

1. Although Par-3 and Par-6 are shown to form a liquid-like phase *in vitro*, it is unclear whether their aggregates are liquid-like in cells. For the mammalian cell culture studies, aggregate fusion events should be assessed for liquid-like properties. Similarly, a small portion of a single aggregate could be photobleached to allow for monitoring of potential recovery from liquid-like flow from other regions of the same aggregate.
2. A recent eLife paper from the Matsuzaki lab showed amorphous aggregates of Par-3 in cell culture. These aggregates seemed non-liquid-like. This paper should be discussed and referenced.
3. The *in vivo* role of the phase separation is based on the correlation that mutations disrupting phase separation *in vitro* also disrupt neuroblast polarization and division. A more definitive test would be evaluating the ability of the unstructured low-complexity domain of FUS (or the PB1 domain of p62) to function in place of the Par-3 oligomerization domain for neuroblast polarization and division.
4. The regulation of the aggregation by aPKC phosphorylation is a potentially interesting point. However, it was not clear if the Par-3 phospho-mimic construct was expressed at the same levels as the control construct. It is possible that the S to E mutations affect the total levels of the protein which would indirectly affect the degree of aggregation.
5. Also, it was unclear why the constitutively active aPKC construct did not affect the aggregates (e.g. by phosphorylating Par-3). Can the authors provide an explanation? The relationship between aPKC activity and the aggregates is unclear. In particular, there is no assay of aPKC activity reported in the study.
6. Further characterization of the specificity of the 1,6-hexanediol treatment is warranted. For example, does the chemical affect the homo-oligomerization of the Par-3 oligomerization domain specifically or of the Par-6 PB1 domain specifically, or does it affect the phase separation by promoting less specific supporting interactions? Effects of the chemical could be further evaluated *in vitro*.

More minor issues:

1. On page 8 line 10, the authors say they use a cell lysate GST pull down to detect a direct interaction. Since the cell lysate contains many components, conclusions about a direct interaction can be difficult.
2. The 3D reconstructions in supplemental figure 1 are difficult to interpret and do not add much beyond what is shown in the corresponding main figure.
3. For Fig S3C, gel lanes seem to be mis-labelled. It seems the first three lanes are inputs.
4. A recent eLife paper from the Prehoda lab described the cell cycle dependent clustering of the

Par proteins in *Drosophila* neuroblasts. It should be referenced.

5. There may be a typo in this sentence: "GST-Par3N could pull down much more Par6 β than GST-Par3N PDZ3 did". What is the difference between "GST-Par3N" and "GST-Par3N PDZ3"?

Reviewer #2 (Remarks to the Author):

This manuscript presents strong evidence that a truncated form of mammalian PAR-3 can form liquid-liquid phase separated aggregates when over-expressed in tissue culture cells and that PAR-6 is incorporated into and enhances the formation of these aggregates. The authors have done a good job in mapping the domains involved in this process (although the N terminal domain of PAR-3 (NTD) was already known to oligomerise) and confirm the previously reported interaction between PDZ3 of PAR-3 and the C-terminus of PAR-6. However, their attempts to test the *in vivo* relevance of these results in *Drosophila* neuroblasts are less convincing. The deletion of the NTD domain of *Drosophila* Par-3 (Bazooka) was already known to disrupt its function and the two other mutants tested, a mutation of PDZ3 and deletion of PAR-6 PB1 could be mislocalized for other reasons, for example because the interaction between PAR-6 and aPKC has been disrupted. Thus, while the idea that the PAR complex forms liquid-liquid phase separated aggregates is very attractive, the manuscript lacks convincing data to prove that these structures form *in vivo*. I don't think that this should necessarily preclude publication, but the conclusions need to be a bit more circumspect as many proteins aggregate when over-expressed but behave differently in their normal context.

Specific points:

"In these NBs, Par proteins and Mira were partially localized with certain cytoplasmic distribution when treated with 5% 1,6-hexanediol, and fully diffused in cytoplasm with 10% 1,6-hexanediol." The signal for Baz, aPKC and PAR-6 looks punctate after hexanediol treatment rather than diffuse, in sharp contrast to Miranda. This suggests that their localization is disrupted, not their clustering.

PAR3N lacks the aPKC binding/phosphorylation site in CR3. It is therefore surprising that aPKC is recruited to the clusters. Can it still interact with the PB1 domain of PAR-6 even when this is oligomerised?

Reviewer #3 (Remarks to the Author):

Liu, Gu, Yang et al. study Par3/Par6/aPKC complex formation using a combination of *in vitro* reconstitution assays, X-ray crystallography and *in vivo* experiments in *Drosophila* neuroblasts. They find that a mutant of Par3 that mimics its open conformation phase separates *in vitro* and forms spherical assemblies *in vivo*. Par6 enhances the phase separation of Par3, whereas aPKC is recruited to the Par3/Par6 phases as a client. The authors propose that Par complex formation during embryonic polarization is driven by phase separation in a cell cycle-dependent manner. Two intriguing findings particularly support this notion: kinase activity of aPKC dissolves Par complex condensates and interfering with Par complex phase separation causes defects in polarization of neuroblasts during asymmetric cell division. Overall, this is an interesting study of potential interest beyond the cell polarization field. However, there are a number of concerns that have to be addressed to make a convincing case, which are outlined in the following. Moreover, I want to strongly encourage the authors to try and do a better job in presenting their findings, both in the main text and the figures.

Major Concerns:

- 1.) Much of the paper rests on the Par3N mutant. However, this construct only comprises approx. half of the protein (Fig. 3a). This begs the question if there is another way of testing the model that auto-inhibition prevents phase separation of full-length Par3. Isn't there a more minimal mutant of Par3 that relieves auto-inhibition? Indeed, it is not clear which parts of the protein the Par3 Δ N12 mutant contains that the authors use to make this point. Is that what Chen et al. 2017 refer to as Par3 Δ 4N1/N2? Or is that just a deletion of the regions annotated 4N 1&2 in Fig.3a? If so, that would be a more minimal mutant of Par3. Why didn't the authors use this construct for their assays? It would help if the authors would provide a cartoon outlining the domain architecture of all constructs used in this study and present this right away in Fig.2.
- 2.) It appears as if Chen et al. 2017 use a similar construct to the Par3N mutant (residues 1-685 as judged by Fig. 3a) used here (Par3 1-712). However, when expressed in the same cells as here (Cos7), they do not observe the formation of puncta, but rather the formation of fibrillar structures. How do the authors explain these differences? How does it affect their interpretation that Par3N undergoes phase separation?
- 3.) The co-expression experiments of the different Par3, Par6 and PKA mutants are especially hard to digest. The authors start by investigating phase separation of the Par complex in Cos7 cells and find that co-expression of Par3N and Par6 β enhances the phase separation phenotype. However, co-expression of Par3N and PKC ζ not only 'did not show such promotion effect' (page 7, lines 19-20), but rather counteracted Par3N puncta formation. Given that (i) the Par3N construct lacks the PKC ζ interaction/phosphorylation site, (ii) phosphorylation of Par3 by aPKC causes complex disassembly and (iii) that it has been suggested that Par6/aPKC form constitutive heterodimers (Rodriguez et al., 2017; Graybill et al., 2012), one immediately wonders about the physiological relevance of both the ideas that Par6 stimulation of Par3 phase separation and that PKC is a Par condensate client. This is only addressed much later in the manuscript and most of the data is unfortunately hidden in the supplementary material. Yet, it remains unclear why the constitutively active PKC ζ mutant is stably recruited to droplets, whereas phospho-mimetic mutants in Par3 Δ 4N12 interfere with droplet formation. Is it possible that PKC ζ is not active in droplets? Or is the PKC ζ A120E mutant in fact not constitutively active? Do the authors see cell cycle-dependent dissolution of Par3 Δ 4N12/Par6b/ PKC ζ droplets in Cos7 cells?
- 4.) The authors went to great length to test the phase properties of their Par assemblies, both in vivo and in vitro. However, a few questions remain, in particular about the material state of the Par complex droplets. For example, it is not clear whether Par proteins really undergo liquid-liquid phase separation (i.e. form condensates with liquid-like properties) or phase separate into hydrogel-like droplets.
 - a. First, it would be very helpful if the authors would relate the FRAP kinetics they observe to the size of the assemblies they bleach. This analysis would allow to (i) make a more convincing point about the exchange dynamics between the Par condensates and the cytoplasm, (ii) estimate the material properties of the Par condensates and (iii) allow for a better comparison between the in vivo and in vitro droplets.
 - b. Second, the data shown in Fig.4a/b does not fully support the conclusion that Par3N/Par6 β droplets undergo fusion, as images of the fully merged droplets are missing. (This may be visible in Supplemental Movies 4/5, which however were not included in the material for the reviewers.) To strengthen this point, it would be helpful to quantify the aspect ratios before/after fusion.
 - c. Third, it is not clear why the authors performed most of their in vitro assays at a concentration of 25 μ M, especially given that they observe droplet formation at much more physiological concentrations. 25 μ M seems absurdly high and the authors themselves caution for artifacts at such high concentrations (page 18, lines 20-21 and page 19, lines 1-12). For example, do the Par3 droplets harden at such high concentrations, as know for e.g. for Fus (Patel et al., 2015), thereby influencing droplet dynamics? Adding to the dissonance is the unfortunate phrasing in the results section. The authors first state that they formed droplets at 25 μ M (page 9, lines 17-19). Then, they go on to say that the amount and size of the droplets increased with increased protein concentration (page 10, lines 10-12). For comparison, it would be extremely helpful if the authors could estimate the endogenous concentration of the Par complex components, as well as estimate their concentration in the Cos7 overexpression system.
- 5.) The authors observe that point mutations in the NTD interfere with Par3N phase separation and

raise the interesting and important question whether Par3 filament formation (via self-association of its NTD) or phase separation underlies Par clustering (page 13, lines 11-13). To differentiate between these models, they replace the NTD with self-associating domains from p62 or FUS. They observe that this restores puncta formation and present this as an argument supporting the idea that phase separation is the driving force. However, the p62 chimera mainly forms helical structures and it is not clear that the remaining amorphous clumps are indeed liquid-like droplets. It is just as likely that these structures are aggregates due to overexpression. Thus, further investigation of the structures is required. Moreover, it is not surprising that the FUS LCD can rescue puncta formation given it is known to drive phase separation, although again, further investigation is required to corroborate this conclusion. Importantly, it is also not clear if the p62 and FUS chimeras can functionally replace the NTD in vivo. In fact, these experiments can only test whether phase separation can promote Par complex clustering in principle, but not provide direct evidence that the Par3 NTD also does so by driving phase separation. This needs to be taken into consideration.

6.) Many of the figures are hard to read, especially for readers with impaired color vision. This pertains especially to Figures 1, 7 and Supplementary Figures 1, 7, 8. However, also the labels in Fig. 5c,d,f are hard to read.

Minor Concerns:

- 1.) Title: It appears that the word 'condensates' is redundant/unnecessary. Perhaps choosing a title that also reflects the functional consequences of the study's findings would be better.
- 2.) Introduction: A sentence introducing PDZ domains is missing. I also suggest removing the specification 'PDZ1' in line 15 on page 3 to set up the problem and then add a sentence that it is unclear through which PDZ domain Par3 bind Par6. Then it will be much easier to understand the details the authors provide on PDZ domains.
- 3.) Introduction: A sentence introducing the CR3 domain is missing.
- 4.) Introduction, page 4, line 4: verb ('are?') missing.
- 5.) Results, page 5, lines 15-19: reference for C. elegans counterparts missing.
- 6.) Results, page 6, line 9: typo: '1,6-hexanediol'
- 7.) Results, page 6, line 10: three papers for the use of hexanediols to disrupt phase separation are cited, however not the original Ribbeck and Görlich paper.
- 8.) Discussion, page 18, lines 11-12: this finding is hidden in Fig.4, but never clearly presented in the results section (see above).
- 9.) Discussion, page 20, lines 20-21: where is that shown? At least a reference is required here. Also see major concerns above.
- 10.) Methods: structure not yet available for review. It would also help to mention the PDB accession code at the end of the 'crystallography' part.

(Our responses to the reviewers' comments are shown in italics and highlighted in blue):

Reviewer #1:

This paper reports the phase separation of Par complex components into a liquid-like state in vitro. Structural and protein interaction studies characterize contributing mechanisms. Studies also reveal that the proteins of interest can form large aggregates with continual turnover in cell culture, and these studies allowed further characterization of the molecular mechanisms contributing to the aggregation (including the oligomerization domain of Par-3, an interaction site between Par-3 and Par-6, and oligomerization of the Par-6 PB1 domain). Par-3 can form the aggregates on its own, but Par-6 is found to enhance their formation. aPKC also joins the aggregates and aPKC phosphorylation of Par-3 seems to reduce its ability to form aggregates. A chemical inhibitor of liquid phase separation disrupts the Par-3-Par-6 phase separation in vitro, and the formation of Par-3 puncta in dividing *Drosophila* neuroblasts. Further suggesting a role for the phase separation in the dividing neuroblasts, alterations to Par-3 or Par-6 that disrupt their phase separation also disrupt their polarization in neuroblasts, as well as the division of the neuroblasts. Overall, this paper presents a compelling case for Par complex phase separation important for asymmetric cell division. This finding would be of substantial interest. However, a number of issues should be addressed.

We thank the reviewer for his/her positive comments on this work.

Major issues:

1) Although Par-3 and Par-6 are shown to form a liquid-like phase in vitro, it is unclear whether their aggregates are liquid-like in cells. For the mammalian cell culture studies, aggregate fusion events should be assessed for liquid-like properties. Similarly, a small portion of a single aggregate could be photobleached to allow for monitoring of potential recovery from liquid-like flow from other regions of the same aggregate.

Following the reviewer's suggestion, we have assessed the liquid-like property of Par3N/Par6 β aggregates in COS7 cells. Par3N and Par6 β -enriched puncta were originally rather small and gradually fused into larger ones in a time-dependent manner (Fig. 2c, presented below for convenience of viewing). In FRAP assay, aggregates with diameters of 2/3/4 μ m were photobleached by laser with a diameter of 2 μ m at 488 nm. Aggregates with diameters of 3/4 μ m had much faster recovery speeds than those with diameters of 2 μ m, indicating that the molecular exchange within the aggregates is faster than that between the aggregate and the dilute cytoplasm (Figure 1). Such liquid-like flow within the same aggregate demonstrates that Par3N/Par6 β aggregates in COS7 cells are indeed liquid-like droplets.

Figure 2c, Representative time-lapse images showing that the GFP-Par3N and mCherry-Par6β-positive puncta undergo time-dependent fusion.

Figure 1. FRAP analysis of the GFP-Par3N signal of GFP-Par3N/Flag-Par6β aggregates with different sizes in COS7 cells. The red/black/blue curve represents the averaged FRAP data of 20 puncta from 10 cells. Time 0 refers to the time point of the photobleaching pulse. All data are represented as mean \pm SD.

2) A recent eLife paper from the Matsuzaki lab showed amorphous aggregates of Par-3 in cell culture. These aggregates seemed non-liquid-like. This paper should be discussed and referenced.

Matsuzaki's data (Kono, eLife, 2019) showed that in S2 cells overexpressed with Par3-GFP, Par-dots and amorphous Par-islands can undergo fusion and fission (Fig. 1Ia&b, presented below for convenience of viewing), displaying a liquid-like property. Matsuzaki also observed the dots-like structure of Par3-GFP in dividing Drosophila neuroblasts (Figure 1Ic), and the same phenomenon was observed in Figure 1 in our manuscript (Fig. 1b). As Par3 oligomerizes both in vitro and in vivo, the amorphous structure of Par-islands (Figure 1Id) in S2 cells may arise from the overexpression and over-accumulation of Par3 in the membrane region. According to a recent study of reconstituted PSD assembly condensates on lipid membrane bilayers (Zeng, Cell, 2018), PSD clusters gradually grew and fused into larger ones with irregular shapes, and finally coalesced into a mesh-like network (Figure 1Ie). Similarly, the overexpressed Par3-GFP in S2 cells may form amorphous Par-islands attached to the inner surface of plasma membrane. Following the reviewer's suggestion, we have referenced this paper and discussed in our revised manuscript (Page 20, Line 1-3; Page 21, Line 3-9).

Figure II a-d, Figures adapted from Matsuzaki's *eLife* paper (Kono, *eLife*, 2019). **a**, Time-lapse imaging of Par6-GFP showing the fusion and fission of Par-dots (arrowheads in the upper panel), and the growth of a Par-dot (arrowheads in the lower panel). Scale bar, 1 μ m. **b**, Time-lapse image of Par-islands visualized by Par6-GFP. Arrowheads indicate dynamic shape changes, fusion (arrowheads, upper panel) and the dissociation of Par-islands (arrowheads, lower panel). Scale bar, 1 μ m. **c**, Localization of the Par3-GFP in a mitotic neuroblast of a *Drosophila* brain expressing Par3-GFP, taken from a third instar larvae and stained for GFP (green) and Miranda (red). Scale bar, 5 μ m. **d**, An image of a S2 cell expressing both myc-Par3-mKate2 and Par6-GFP, immunostained for myc. Scale bar, 5 μ m. **e**, Figures adapted from another LLPS paper (Zeng, *Cell*, 2018). Time-lapse confocal images showing that homogeneously distributed Cy3-His-NR2B on membrane bilayers gradually clustered upon addition of the rest of five unlabeled PSD proteins. Scale bar, 2 μ m.

3) The *in vivo* role of the phase separation is based on the correlation that mutations disrupting phase separation *in vitro* also disrupt neuroblast polarization and division. A more definitive test would be evaluating the ability of the unstructured low-complexity domain of FUS (or the PB1 domain of p62) to function in place of the Par-3 oligomerization domain for neuroblast polarization and division.

*We thank the reviewer for noting this issue for us and we have generated GFP knock-in Drosophila strains expressing endogenous levels of the fluorescently tagged Baz, Baz Δ NTD mutant, and FUS_S-Baz chimera mutant (Baz NTD was replaced by FUS 1-141). In sharp contrast to the rescue assay results that Baz Δ NTD mutant lost its apical condensation in about 80% NBs (Fig. 7d&e), the knock-in mutant showed obvious apical condensation in all NBs (Fig. 9a-d, presented below for convenience of viewing). As LLPS depends on concentration of proteins, we assume that the distinct phenotypes of the same mutant likely arise from the protein concentration difference in two assays as it was reported that over-expressing Baz NBs resulted in ectopic localization (Petronczki, *Nat Cell Biol*, 2001). However, in GFP-Baz Δ NTD knock-in flies, we could observe that both Baz and the basal protein Mira showed significant cytoplasmic diffusion, demonstrating the essential role of NTD in efficient protein local condensation. Importantly, the FUS_S-Baz mutant largely restored the crescent localization of apical and basal proteins, and largely rescued the smaller brain phenotype of the NTD deletion mutant. The above data strongly indicate that Par3 NTD-mediated Par complex condensation is critical from polarized Par complex clustering in fly NBs.*

Fig. 9 Par3 NTD-mediated Par complex condensation is critical from polarized Par complex clustering in fly NBs. a, Knock-in larval NBs expressing GFP-tagged Baz WT, Baz Δ NTD, or FUS_S-Baz chimera mutant with staining of endogenous Par6, aPKC, Mira at metaphase. ToPro-3 in blue. Scale bars, 5 μm . b, Statistical data for (a). c, Larval brains expressing GFP-tagged baz wt, baz Δ NTD or FUS_S-baz chimera mutant. d, Statistical data for (c). e, Representative time-lapse FRAP images showing that GFP-Baz WT signal within the crescent formed by endogenous GFP-Baz WT recovered within a few minutes.

4) The regulation of the aggregation by aPKC phosphorylation is a potentially interesting point. However, it was not clear if the Par-3 phospho-mimic construct was expressed at the same levels as the control construct. It is possible that the S to E mutations affect the total levels of the protein which would indirectly affect the degree of aggregation.

We thank the reviewer for noting this issue for us. We have checked the protein expression levels, and Par3 Δ 4N12 phospho-mimic construct was expressed at a slightly higher level than the control construct (Supplementary Figure 5d, presented below for convenience of viewing), supporting that aPKC-mediate Par3 phosphorylation may impair Par3/Par6 LLPS property.

Supplementary Figure 5d, Expression of Par3N WT and various mutants with Par6β in COS7 cells in (a) were detected by Western-blotting assay.

5) Also, it was unclear why the constitutively active aPKC construct did not affect the aggregates (e.g. by phosphorylating Par-3). Can the authors provide an explanation? The relationship between aPKC activity and the aggregates is unclear. In particular, there is no assay of aPKC activity reported in the study.

Following this reviewer's suggestion, we conducted a sedimentation assay of Par3 1-854 (containing the aPKC phosphorylation sites) and Par6β with or without incubation of flag-PKCι purified from HEK293 cells. According to Spitaler's results, the PKCι A120E mutant is constitutively active in the absence of Par3 and Par6β (Spitaler, JBC, 2000). Both PKCι WT and A120E efficiently phosphorylated Par3 1-854 (indicated by the upshifted phos-tag band) in the supernatant fraction but not in the Par3/Par6β/PKCι condensates (Figure 6f), further supporting our model that though PKCι could be recruited into the Par3/Par6β condensates, its activity is suppressed. We thus propose that aPKC could be recruited into the Par3/Par6 condensates as an inactive client, and then transferred to the apical membrane region. The apically localized aPKC may be activated by cell cycle dependent factors such as CDC42, leading to Par3 phosphorylation by aPKC and dissociation of Par3 from Par6/aPKC complex. The activated aPKC could then phosphorylate adaptors and fate determinants proteins, leading to their basal localization.

Figure 6f. Sedimentation assay showing the distribution of proteins between aqueous-solution/supernatant (S) and condensed liquid phase/pellet (P) fractions. Both PKCι

WT and A120E mutant phosphorylated Par3 1-854 in the supernatant fraction but not in the pellet fraction. The band of phosphorylated Par3 1-854 was resolved by Phos-tag PAGE.

6) Further characterization of the specificity of the 1,6-hexanediol treatment is warranted. For example, does the chemical affect the homo-oligomerization of the Par-3 oligomerization domain specifically or of the Par-6 PB1 domain specifically, or does it affect the phase separation by promoting less specific supporting interactions? Effects of the chemical could be further evaluated in vitro.

We thank the reviewer for noting this issue for us. 1,6-hexanediol is known to disrupt hydrophobic interaction-induced LLPS. In line with the Par3 PDZ3/Par6 β PBM complex structure, fluorescence polarization assay revealed that 1,6-hexanediol impaired the hydrophobic interaction between Par3 PDZ3 and Par6 β PBM peptide in a dose-dependent manner (Supplementary Figure 3e). We thus assume that 1,6-hexanediol-induced disruption of Par3N/Par6 β interaction leads to dispersion of Par3N/Par6 β condensates.

Figure 3e. Fluorescence polarization-based measurements of the binding affinities between Par3 PDZ3 and Par6 β PBM peptide with various concentrations of 1,6-hexanediol.

Minor issues:

1) On page 8 line 10, the authors say they use a cell lysate GST pull down to detect a direct interaction. Since the cell lysate contains many components, conclusions about a direct interaction can be difficult.

Thanks for the reviewer's advice and we have modified our description (Page 8, Line 6).

2) The 3D reconstructions in supplemental figure 1 are difficult to interpret and do not add much beyond what is shown in the corresponding main figure.

We have removed the 3D reconstructions data.

3) For Fig S3C, gel lanes seem to be mis-labelled. It seems the first three lanes are inputs.

Thanks for the reviewer's advice and we have corrected the label in Fig S2c (presented below for convenience of viewing).

Supplementary Figure 2c. GST pull-down assay showing that GST-tagged Par6 β PB1 bound to MBP-tagged Par6 β PB1.

4) A recent eLife paper from the Prehoda lab described the cell cycle dependent clustering of the Par proteins in *Drosophila* neuroblasts. It should be referenced.

We thank the reviewer for noting this issue for us and we have referenced the paper in our revised manuscript (Page 20, Line 1-3).

5) There may be a typo in this sentence: “GST-Par3N could pull down much more Par6 β than GST-Par3N PDZ3 did” . What is the difference between “GST-Par3N” and “GST-Par3N PDZ3” ?

GST-Par3N stands for NTD plus PDZ1-3, and Par3N PDZ3 only stands for PDZ3. We have summarized various protein constructs in Supplementary Table 1.

Reviewer #2:

This manuscript presents strong evidence that a truncated form of mammalian PAR-3 can form liquid-liquid phase separated aggregates when over-expressed in tissue culture cells and that PAR-6 is incorporated into and enhances the formation of these aggregates. The authors have done a good job in mapping the domains involved in this process (although the N terminal domain of PAR-3 (NTD) was already known to oligomerise) and confirm the previously reported interaction between PDZ3 of PAR-3 and the C-terminus of PAR-6. However, their attempts to test the in vivo relevance of these results in Drosophila neuroblasts are less convincing. The deletion of the NTD domain of Drosophila Par-3 (Bazooka) was already known to disrupt its function and the two other mutants tested, a mutation of PDZ3 and deletion of PAR-6 PB1 could be mislocalized for other reasons, for example because the interaction between PAR-6 and aPKC has been disrupted. Thus, while the idea that the PAR complex forms liquid-liquid phase separated aggregates is very attractive, the manuscript lacks convincing data to prove that these structures form in vivo. I don't think that this should necessarily preclude publication, but the conclusions need to be a bit more circumspect as many proteins aggregate when over-expressed but behave differently in their normal context.

We thank the reviewer for his/her positive comments on this work.

Specific points:

1) “In these NBs, Par proteins and Mira were partially localized with certain cytoplasmic distribution when treated with 5% 1,6-hexanediol, and fully diffused in cytoplasm with 10% 1,6-hexanediol.” The signal for Baz, aPKC and PAR-6 looks punctate after hexanediol treatment rather than diffuse, in sharp contrast to Miranda. This suggests that their localization is disrupted, not their clustering.

We thank the reviewer for noting this issue for us. The puncta-like signals were actually noise. We have updated the images in Supplementary Figure 1 (presented below for convenience of viewing).

Supplementary Figure 1. Phase separation of the Par complex was reversible in vivo. *a*, In wild-type NBs treated with 5% 1,6-hexanediol, some NBs exhibited cytoplasmic localization of Baz (in 20% of NBs), Par6 (in 15% of NBs), aPKC (in 20% of NBs), and Mira (in 15% of NBs). When treated with 10% 1,6-hexanediol, virtually all NBs showed cytoplasmic localization of these proteins. *b*, After removal of 1,6-hexanediol, the apical distribution of Baz, Par6 and aPKC and basal localization of Mira were restored in all NBs examined. Red arrowheads point to the apical cortex whereas white ones indicate the basal cortex. ToPro-3 in blue. Scale bars, 5 μ m. *c*, Statistical data for (a) and (b).

2) PAR3N lacks the aPKC binding/phosphorylation site in CR3. It is therefore surprising that aPKC is recruited to the clusters. Can it still interact with the PB1 domain of PAR-6 even when this is oligomerised?

aPKC is recruited into the Par3N/Par6 β condensates through Par6 β . GST pull-down data confirmed that PKC ζ PB1 can still interact with the PB1 domain of Par6 β even though it can self-oligomerize (Figure III).

Figure III. GST pull-down assay by Coomassie Blue staining showing that GST-PKC ζ PB1 bound to Trx-Par6 β PB1.

Reviewer #3:

Liu, Gu, Yang et al. study Par3/Par6/aPKC complex formation using a combination of in vitro reconstitution assays, X-ray crystallography and in vivo experiments in *Drosophila* neuroblasts. They find that a mutant of Par3 that mimics its open conformation phase separates in vitro and forms spherical assemblies in vivo. Par6 enhances the phase separation of Par3, whereas aPKC is recruited to the Par3/Par6 phases as a client. The authors propose that Par complex formation during embryonic polarization is driven by phase separation in a cell cycle-dependent manner. Two intriguing findings particularly support this notion: kinase activity of aPKC dissolves Par complex condensates and interfering with Par complex phase separation causes defects in polarization of neuroblasts during asymmetric cell division. Overall, this is an interesting study of potential interest beyond the cell polarization field. However, there are a number of concerns that have to be addressed to make a convincing case, which are outlined in the following. Moreover, I want to strongly encourage the authors to try and do a better job in presenting their findings, both in the main text and the figures.

We thank the reviewer for his/her positive comments on this work.

Major concerns:

1) Much of the paper rests on the Par3N mutant. However, this construct only comprises approx. half of the protein (Fig. 3a). This begs the question if there is another way of testing the model that auto-inhibition prevents phase separation of full-length Par3. Isn't there a more minimal mutant of Par3 that relieves auto-inhibition? Indeed, it is not clear which parts of the protein the Par3 $\Delta 4N12$ mutant contains that the authors use to make this point. Is that what Chen et al. 2017 refer to as Par3 $\Delta 4N1/N2$? Or is that just a deletion of the regions annotated 4N 1&2 in Fig.3a? If so, that would be a more minimal mutant of Par3. Why didn't the authors use this construct for their assays? It would help if the authors would provide a cartoon outlining the domain architecture of all constructs used in this study and present this right away in Fig.2.

We thank the reviewer for noting this issue for us and we have listed all the constructs used in this study in Supplementary Table 1. So far as we know, Par3 $\Delta 4N12$ construct is with the minimal deletion that can relieve the auto-inhibition of Par3, and this construct is the same one in Chen's paper (Chen, Dev Cell, 2013) referred to as Par3 $\Delta 4N1/N2$. However, the expression level and protein behavior of recombinant Par3 $\Delta 4N12$ was not suitable for in vitro LLPS experiments (Figure IV). As Par3N exhibited a comparable ability to form puncta in COS7 cells as Par3 $\Delta 4N12$, we thus used Par3N in the LLPS assay.

Figure IV. Purification of recombinant Par3 Δ 4N12 (highlighted in red rectangle) detected by SDS-PAGE.

2) It appears as if Chen et al. 2017 use a similar construct to the Par3N mutant (residues 1-685 as judged by Fig. 3a) used here (Par3 1-712). However, when expressed in the same cells as here (Cos7), they do not observe the formation of puncta, but rather the formation of fibrillar structures. How do the authors explain these differences? How does it affect their interpretation that Par3N undergoes phase separation?

In addition to puncta formation, we also observed fibrillar structures of Par3N in COS7 (Figure V, presented below for convenience of viewing). The ratio of puncta vs fibrillar structures is about 1:4. We have discussed with the authors of the paper (Chen, Dev Cell, 2013) and they also observed puncta formation in some cases. We assume that Par3N may exist in two pools and we focused on the puncta form of the protein in this study.

Figure V. We observed both puncta and fibrillar structures of GFP-Par3N in COS7 cells.

3) The co-expression experiments of the different Par3, Par6 and PKA mutants are especially hard to digest. The authors start by investigating phase separation of the Par complex in Cos7 cells and find that co-expression of Par3N and Par6 β enhances the phase separation phenotype. However, co-expression of Par3N and PKC ζ not only ‘did not show such promotion effect’ (page 7, lines 19-20), but rather counteracted Par3N puncta formation. Given that (i) the Par3N construct lacks the PKC ζ interaction/phosphorylation site, (ii) phosphorylation of Par3 by aPKC causes complex disassembly and (iii) that it has been suggested that Par6/aPKC form constitutive heterodimers (Rodriguez et al., 2017; Graybill et al., 2012), one immediately wonders about the physiological relevance of both the ideas that Par6 stimulation of Par3 phase separation and that PKC is a Par condensate client. This is only addressed much later in the manuscript and most of the data is unfortunately hidden in the supplementary material. Yet, it remains unclear why the constitutively active PKC ζ mutant is stably recruited to droplets, whereas phospho-mimetic mutants in Par3 Δ 4N12 interfere with droplet formation. Is it possible that PKC ζ is not active in droplets? Or is the PKC ζ A120E mutant in fact not constitutively active? Do the authors see cell cycle-dependent dissolution of Par3 Δ 4N12/Par6 β / PKC ζ droplets in Cos7 cells?

Our manuscript began with the observation of Par complex clustering in vivo, then we observed the promotion of Par6 β in Par3N puncta formation in living cells and thus

characterized the underlying molecular mechanism, then we analyzed the effect of aPKC in Par condensates formation, and finally validated the physiological relevance of Par LLPS during ACD of NBs. Following the reviewer's suggestion, we have moved some PKC data into the main figure (Figure 6).

Co-expression of Par6 β , PKC ζ , or control mCherry with Par3N would somehow decrease the expression level of Par3N. As the LLPS of Par3N is concentration dependent, a low protein level of Par3N may lead to failure of puncta formation. In sharp contrast, coexpression of Par6 β and Par3N significantly promoted the puncta formation of Par3N, pointing to a positive role of Par6 β in Par3N LLPS.

To test whether Par3/Par6 β condensates-embedded aPKC is active or not, we conducted a sedimentation assay of Par3 1-854 (containing the aPKC phosphorylation sites) and Par6 β with or without incubation of Flag-PKC ζ purified from HEK293 cells. According to Spitaler's results, the PKC ζ A120E mutant is constitutively active in the absence of Par3 and Par6 β (Spitaler, JBC, 2000). Both PKC ζ WT and A120E efficiently phosphorylated Par3 1-854 (indicated by the upshifted phos-tag band) in the supernatant fraction but not in the Par3/Par6 β /PKC ζ condensates (Figure 6f), further supporting our model that though PKC ζ could be recruited into the Par3/Par6 β condensates, its activity is suppressed. We thus propose that aPKC could be recruited into the Par3/Par6 condensates as an inactive client, and then transferred to the apical membrane region. The apically localized aPKC may be activated by cell cycle dependent factors such as CDC42, leading to Par3 phosphorylation by aPKC and dissociation of Par3 from Par6/aPKC complex. The activated aPKC could then phosphorylate adaptors and fate determinants proteins, leading to their basal localization.

Figure 6f. Sedimentation assay showing the distribution of proteins between aqueous-solution/supernatant (S) and condensed liquid phase/pellet (P) fractions. Both PKC ζ WT and A120E mutant phosphorylated Par3 1-854 in the supernatant fraction but not in the pellet fraction. The band of phosphorylated Par3 1-854 was resolved by Phos-tag PAGE.

We could not observe cell cycle-dependent dissolution of Par3 Δ 4N12/Par6 β /PKC ζ in COS7 cells, possibly due to the difference between COS7 cells and neuroblasts, and COS7 cells are overexpression systems.

4) The authors went to great length to test the phase properties of their Par assemblies, both in vivo and in vitro. However, a few questions remain, in particular about the material state of the Par complex droplets. For example, it is not clear whether Par proteins really undergo liquid-liquid phase separation (i.e. form condensates with liquid-like properties) or phase

separate into hydrogel-like droplets.

a. First, it would be very helpful if the authors would relate the FRAP kinetics they observe to the size of the assemblies they bleach. This analysis would allow to (i) make a more convincing point about the exchange dynamics between the Par condensates and the cytoplasm, (ii) estimate the material properties of the Par condensates and (iii) allow for a better comparison between the in vivo and in vitro droplets.

We thank the reviewer for noting this issue for us and we have redone the FRAP assay in vitro. Par3N/Par6β droplets with various sizes were photobleached by a laser with a diameter of 2 μm. Larger droplets had faster recovery rates than small ones, implying that the internal mobility within the aggregates is more dynamic than that between the aggregate and the cytoplasm (Figure VIa). Such liquid-like flow within the same aggregate demonstrates that Par3N/Par6β droplets are indeed liquid-like phase. However, we found that Par3N/Par6β droplets transition into gel-like structures over time. The droplets rarely fused together after 20 min of mixing two proteins (Figure VIb).

Figure VI. Par3N/Par6β droplets are condensed liquid-like phase. a, FRAP analysis of the droplets formed by mixing iFluor488-labeled Par3N with Par6β. The curves represent FRAP recovery curves of iFluor™ 488-Par3N by averaging signals of 15 droplets with different sizes each after photobleaching. Time 0 refers to the time point of the photobleaching pulse. Data are represented as mean ± SD. b, Fusion assay of the droplets formed by mixing iFluor488-Par3N with Cy3-Par6β.

b. Second, the data shown in Fig.4a/b does not fully support the conclusion that Par3N/Par6β droplets undergo fusion, as images of the fully merged droplets are missing. (This may be visible in Supplemental Movies 4/5, which however were not included in the material for the reviewers.) To strengthen this point, it would be helpful to quantify the aspect ratios before/after fusion.

As mentioned in the above point, Par3N/Par6β droplets transition into gel-like structures over time. However, we indeed observed fully merged droplets in some cases (Figure VII).

Figure VII. Fusion assay of the droplets formed by mixing iFluor488-Par3N with Par6 β . Droplets formed by Par3N/ Par6 β can fully merge.

c. Third, it is not clear why the authors performed most of their in vitro assays at a concentration of 25 μ M, especially given that they observe droplet formation at much more physiological concentrations. 25 μ M seems absurdly high and the authors themselves caution for artifacts at such high concentrations (page 18, lines 20-21 and page 19, lines 1-12). For example, do the Par3 droplets harden at such high concentrations, as know for e.g. for Fus (Patel et al., 2015), thereby influencing droplet dynamics? Adding to the dissonance is the unfortunate phrasing in the results section. The authors first state that they formed droplets at 25 μ M (page 9, lines 17-19). Then, they go on to say that the amount and size of the droplets increased with increased protein concentration (page 10, lines 10-12). For comparison, it would be extremely helpful if the authors could estimate the endogenous concentration of the Par complex components, as well as estimate their concentration in the Cos7 overexpression system.

We indeed observed Par3N/Par6 β droplets formation at physiological concentration in vitro (~ 1 μ M, Figure 4d). However, these droplets were very small and distributed sparsely. So the fusion process of Par3N/Par6 β droplets at this concentration was difficult to be observed. Furthermore, droplets were too small to conduct FRAP assay. We then chose 25 μ M to perform fusion and FRAP analysis.

We have measured the concentrations of overexpressed Par3 in COS7 cells (5.5 μ M) and endogenously expressed Baz in NBs (0.15 μ M; Supplementary Figure 8, presented below for convenience of viewing). The overall protein level of overexpressed Par3 (in Cos7 cells) was high enough to undergo LLPS, that's why Par puncta were observed in the cytoplasm. Whereas the endogenous Baz level (in NBs) was not high enough to lead to its phase separation in the cytoplasm. Compared with the cytoplasmic fraction, we assumed that attachment of Par complex to the two-dimensional membrane led to its enrichment and subsequent LLPS of the proteins in NBs.

Supplementary Figure 8. Quantification of protein concentrations in cells. a, Concentrations of Par3N overexpressed in COS7 cells. The right panel is calibration curve showing the relationship between GFP concentration and GFP fluorescence intensity. The black curve shows the linear regression analysis of the fluorescence intensities versus protein concentrations. b, Concentrations of endogenous Baz in NBs. The right panel is calibration curve showing the relationship between GFP concentration and GFP fluorescence intensity. The black curve shows the linear regression analysis of the fluorescence intensities versus protein concentrations.

5) The authors observe that point mutations in the NTD interfere with Par3N phase separation and raise the interesting and important question whether Par3 filament formation (via self-association of its NTD) or phase separation underlies Par clustering (page 13, lines 11-13). To differentiate between these models, they replace the NTD with self-associating domains from p62 or FUS. They observe that this restores puncta formation and present this as an argument supporting the idea that phase separation is the driving force. However, the p62 chimera mainly forms helical structures and it is not clear that the remaining amorphous clumps are indeed liquid-like droplets. It is just as likely that these structures are aggregates due to overexpression. Thus, further investigation of the structures is required. Moreover, it is not surprising that the FUS LCD can rescue puncta formation given it is known to drive phase separation, although again, further investigation is required to corroborate this conclusion. Importantly, it is also not clear if the p62 and FUS chimeras can functionally replace the NTD in vivo. In fact, these experiments can only test whether phase separation can promote Par complex clustering in principle, but not provide direct evidence that the Par3 NTD also does so by driving phase separation. This needs to be taken into consideration.

We thank the reviewer for noting this issue for us and we have generated GFP knock-in Drosophila strains expressing endogenous levels of the fluorescently tagged Baz, Baz Δ NTD

mutant, and *FUS₅-Baz* chimera mutant (*Baz* NTD was replaced by *FUS* 1-141). In sharp contrast to the rescue assay results that *Baz* Δ NTD mutant lost its apical condensation in about 80% NBs (Fig. 7d&e), the knock-in mutant showed obvious apical condensation in all NBs (Fig. 9a-d, presented below for convenience of viewing). As LLPS depends on concentration of proteins, we assume that the distinct phenotypes of the same mutant likely arise from the protein concentration difference in two assays as it was reported that over-expressing *Baz* NBs resulted in ectopic localization (Petronczki, Nat Cell Biol, 2001). However, in GFP-*Baz* Δ NTD knock-in flies, we could observe that both *Baz* and the basal protein *Mira* showed significant cytoplasmic diffusion, demonstrating the essential role of NTD in efficient protein local condensation. Importantly, the *FUS₅-Baz* mutant largely restored the crescent localization of apical and basal proteins, and largely rescued the smaller brain phenotype of the NTD deletion mutant. The above data strongly indicate that *Par3* NTD-mediated *Par* complex condensation is critical from polarized *Par* complex clustering in fly NBs. To simplify the results, we removed the p62 data in this revised manuscript.

Fig. 9 NTD-mediated *Par* clustering relies on its oligomerization-dependent LLPS behavior. *a*, Knockin larval NBs expressing GFP-tagged *Baz* WT, *Baz* Δ NTD, or *FUS₅-Baz* chimera mutant with staining of endogenous *Par6*, *aPKC*, *Mira* at metaphase. ToPro-3 in blue. Scale bars, 5 μm . *b*, Statistical data for (a). *c*, Larval brains expressing GFP-tagged *Baz* WT, *Baz* Δ NTD or *FUS₅-Baz* chimera mutant. *d*, Statistical data for (c). *e*, Representative time-lapse FRAP images showing that GFP-*Baz* WT signal within the crescent formed by endogenous

GFP-Baz WT recovered within a few minutes.

6) Many of the figures are hard to read, especially for readers with impaired color vision. This pertains especially to Figures 1, 7 and Supplementary Figures 1, 7, 8. However, also the labels in Fig. 5c,d,f are hard to read.

Thank you for your advice and we have modified the figures accordingly.

Minor concerns:

1) Title: It appears that the word ‘condensates’ is redundant/unnecessary. Perhaps choosing a title that also reflects the functional consequences of the study’s findings would be better.

Thanks for your advice and we have changed the title as “Phase separation mediated Par complex cluster formation in cell polarity”.

2) Introduction: A sentence introducing PDZ domains is missing. I also suggest removing the specification ‘PDZ1’ in line 15 on page 3 to set up the problem and then add a sentence that it is unclear through which PDZ domain Par3 binds to Par6. Then it will be much easier to understand the details the authors provide on PDZ domains.

We thank for the reviewer’s advice and have modified the text: “Par3 has three PDZ (PSD-95, DLG and ZO-1) domains that mediate protein-protein interactions.”(page3, line14); “Mammalian Par3 was reported to interact with Par6 through its PDZ domain, though it is unclear through which PDZ domain Par3 binds to Par6.” (Page 3, Line 14-18).

3) Introduction: A sentence introducing the CR3 domain is missing.

We thank for the reviewer’s advice and have modified the text: “aPKC, which forms a stable subcomplex with Par6 through their PBI domains, binds to Par3 conserved region 3 (CR3) through its kinase domain” (Page3, Line 20-23).

4) Introduction, page 4, line 4: verb (‘are’?) missing

We have modified the text.

5) Results, page 5, lines 15-19: reference for C. elegans counterparts missing.

We have added the references.

6) Results, page 6, line 9: typo: ‘1,6-hexanediol’

We have modified the text.

7) Results, page 6, line 10: three papers for the use of hexanediols to disrupt phase separation are cited, however not the original Ribbeck and Görlich paper.

We have added the reference following the reviewer’s advice.

8) Discussion, page 18, lines 11-12: this finding is hidden in Fig.4, but never clearly presented in the results section (see above).

We have modified the text.

9) Discussion, page 20, lines 20-21: where is that shown? At least a reference is required here. Also see major concerns above.

We have added the references.

10) Methods: structure not yet available for review. It would also help to mention the PDB accession code at the end of the 'crystallography' part.

We have added the PDB code in the Methods.

Reviewers' comments:

Reviewer #1 (Remarks to the Author):

My past concerns have been addressed effectively. The revised paper carefully describes an interesting and important study that should be of general interest.

Reviewer #2 (Remarks to the Author):

The authors have done a good job at addressing the questions raised by the referees by adding substantial additional data. Although the *in vivo* experiments are not totally conclusive, they make a stronger case for the role of LLPS in the neuroblast and I now feel that the manuscript is more balanced. I recommend that it be accepted without further revisions.

Reviewer #3 (Remarks to the Author):

Liu, Gu, Yang et al. have made major revision to their manuscript in which they suggest that the Par complex is concentrated beneath the apical membrane of neuroblasts (NBs) by a phase separation mechanism. However, even though the manuscript shows that certain Par3 mutants can phase separate together with Par6 *in vitro* and in Cos7 cells, especially the new data presented in Figure 9 adds to previous concerns whether Par3/6 condensate formation is truly necessary (let alone sufficient) to establish asymmetry in NBs.

Specifically, the phase separation/oligomerization-deficient Par3/Baz Δ NTD mutant still shows apical localization at endogenous expression levels in NBs, remains viable and only has mild brain phenotypes that closely resemble the defects observed for the FUS-Par3/Baz chimeras presented in the manuscript as rescue constructs. In addition to these concerns, the previously noted choice of constructs and lack of consistency in use of those constructs in different assays, as well as the suboptimal presentation of the data with illegible and poorly explained figure labels do not help to support the Par condensate model of the authors.

In conclusion, the manuscript is not ready for publication.

(Our responses to the reviewers' comments are shown in italics and highlighted in blue):

Reviewer #1:

My past concerns have been addressed effectively. The revised paper carefully describes an interesting and important study that should be of general interest.

We thank the reviewer for his/her final approval.

Reviewer #2:

The authors have done a good job at addressing the questions raised by the referees by adding substantial additional data. Although the *in vivo* experiments are not totally conclusive, they make a stronger case for the role of LLPS in the neuroblast and I now feel that the manuscript is more balanced. I recommend that it be accepted without further revisions.

We thank the reviewer for his/her final approval.

Reviewer #3:

Liu, Gu, Yang et al. have made major revision to their manuscript in which they suggest that the Par complex is concentrated beneath the apical membrane of neuroblasts (NBs) by a phase separation mechanism. However, even though the manuscript shows that certain Par3 mutants can phase separate together with Par6 *in vitro* and in Cos7 cells, especially the new data presented in Figure 9 adds to previous concerns whether Par3/6 condensate formation is truly necessary (let alone sufficient) to establish asymmetry in NBs.

1) Specifically, the phase separation/oligomerization-deficient Par3/Baz Δ NTD mutant still shows apical localization at endogenous expression levels in NBs, remains viable and only has mild brain phenotypes that closely resemble the defects observed for the FUS-Par3/Baz chimeras presented in the manuscript as rescue constructs.

We thank the reviewer for noting this issue for us. As NTD oligomerization was only one driven force for Par3/Baz LLPS (Fig. 5a-d), we found that the oligomerization-deficient Par3/Baz Δ NTD had impaired but not completely disrupted LLPS property. The Par3N NTDmu (V13D,D70K) mutant could still undergo LLPS with or without Par6 β at higher concentrations (Fig. 1, presented below for convenience of viewing) in solution, suggesting that NTD oligomerization is important but not absolutely required for the multivalency-dependent Par3/Par6 LLPS. Consistent with this data, Baz Δ NTD with endogenous expressing level showed obvious apical condensation in all NBs of GFP-Baz Δ NTD knock-in flies, but with a significant amount of cytoplasmic diffusion (Fig. 9a-b). The similar phenomena were observed for endogenous Par6, aPKC and the basal protein Mira, implying that NTD-mediated LLPS of Baz at physiological condition indeed facilitated the efficient Baz apical localization and thus regulated local condensation of other proteins. Such diffusion effect led to a smaller brain phenotype of flies compared to the wild type strains with statistical significance (Fig. 9c-d). Importantly, the FUS_S-Baz mutant largely restored the crescent localization of apical and basal proteins, and largely rescued the smaller brain phenotype of the NTD deletion mutant (with statistical significance), demonstrating that it is the LLPS ability of NTD facilitates Par complex local condensation. Note that FUS_S was a

small *FUS* fragment with a compromised LLPS property compared to *FUS_L*, and replacing NTD with *FUS*s could only partially rescue the LLPS efficiency of *Par3N* when coexpressed with *Par6* in *COS7* cells (Fig.5e-g). However, *FUS_L-Baz* flies was lethal and thus excluded from this study.

We propose that *Baz/Par3* localization is a balanced result of apical anchoring and LLPS-mediated local condensation (via multivalent protein-protein interaction, self-association, protein-membrane interaction, etc.). Thus, for the knock-in mutant *Baz ΔNTD*, partially impaired LLPS ability led to its less condensed localization and significant cytoplasmic diffusion. However, the situation was different for the overexpressed *Baz NTDmu* (driven by *UAS/GAL4*) in the rescue assay. As LLPS is very sensitive to concentrations of biological components, an overexpression of *Par* proteins especially *Baz/Par3*, the core driving factor of LLPS, may cause artificial promotion of the *Par* complex condensation via LLPS. Whereas the apical anchoring capacity of NBs seems to have a limitation. In *UAS/GAL4*-based rescue assay, the overexpressed *Baz WT* phase condensates may just have reached the threshold of apical anchoring capacity, whereas the LLPS deficient, overexpressed *Baz NTDmu* broke the balance and led to its cortical and cytoplasmic diffusion. If the expression level goes higher, even *Baz WT* can not be afforded apically. Consistent with this notion, high *Flag-Baz* expression in a *WT* background (driven by *insc-gal4*), has a dominant-negative effect and leads to ectopic localization of endogenous *Par* complex throughout the cortex, and consequently disrupts localization of basal proteins (Supplementary Fig. 7c,d). Similarly, ectopic *Baz* localization was observed when exogenous *Baz* is forcedly expressed in embryonic NBs[1, 2].

Fig. 1. Representative SDS-PAGE analysis and quantification data showing the distribution of proteins between aqueous-solution/supernatant (S) and condensed liquid phase/pellet (P) fractions. *Par3N NTDmu* and *Par6β* were mixed at a 1:0 (a) or 1:1 (b) molar ratio at various concentrations. Data represent the results from three independent batches of experiments and are expressed as mean \pm SD.

References:

- Petronczki, M. and J.A. Knoblich, *DmPAR-6 directs epithelial polarity and asymmetric cell division of neuroblasts in Drosophila*. Nat Cell Biol, 2001. **3**(1): p. 43-9.
- Izumi, Y., et al., *Differential functions of G protein and Baz-aPKC signaling pathways in Drosophila neuroblast asymmetric division*. J Cell Biol, 2004. **164**(5): p. 729-38.

2) In addition to these concerns, the previously noted choice of constructs and lack of consistency in use of those constructs in different assays, as well as the suboptimal presentation of the data with illegible and poorly explained figure labels do not help to support the Par condensate model of the authors.

In conclusion, the manuscript is not ready for publication.

As we mentioned in previous response, compared with the Par3N construct, we indeed have a longer fragment Par3 Δ 4N12 which can relieve the auto-inhibition of Par3. However, the expression level and protein behavior of recombinant Par3 Δ 4N12 was too bad to carry on in vitro LLPS experiments. As Par3N exhibited a comparable ability to form puncta in COS7 cells as Par3 Δ 4N12, we thus used Par3N in the LLPS assay.

Due to the poor protein behavior of Par3 Δ 4N12, we used Par3N as a template for the constructs in in vitro LLPS assay and COS7 assay, and these two assays demonstrated the LLPS properties of Par3N/Par6 complex. In transgenic fly rescue assay, full-length Baz was used as a template for all the mutations to preclude the potential side effects induced by other domain truncation. Nevertheless, mutations used in the fly rescue assay were consistent with those in COS7 assays, and the results supported our model that Par3(Baz)/Par6 LLPS led to their apical condensation in NBs. We further made knock-in flies to validate the functional effect of Baz LLPS at its physiological concentration. So far as we know, the physiological function of NTD seems only for the oligomerization of Par3/Baz. Thus, for practical reasons, to specifically probe the consequence of Par LLPS without disturbing its protein-protein interactions, we chose to generate GFP-tagged Baz, Baz Δ NTD mutant, and FUS_S-Baz chimera mutant, and the results further were in line with our model.

Following the reviewer's suggestion, we had tried to provide cartoons outlining the domain architecture of all constructs used in this study in Fig.2 (presented below for convenience of viewing). However, the figure would be too messy. We had listed all the constructs with cartoons in Supplementary Table 1, and further clarified the labels in figure legends. We have also carefully modified the manuscript to clarify our data.

(Our responses to the reviewers' comments are shown in italics and highlighted in blue):

REVIEWERS' COMMENTS

We thank the reviewer for his/her final approval.